# Genome-wide antibiotic-CRISPRi profiling identifies LiaR activation as a strategy to resensitize fluoroquinolone-resistant *Streptococcus pneumoniae*

Bevika Sewgoolam ®, Kin Ki Jim ®, Vincent de Bakker ®, Florian P. Bock ®, Paddy S. Gibson ® & Jan-Willem Veening ® ✉

*Streptococcus pneumoniae* is a human pathogen that has become increasingly resistant to synthetic fluoroquinolone antibiotics that target bacterial topoisomerases. To identify pathways essential under fluoroquinolone stress and potential novel targets to revitalize use of this antibiotic class, we perform genome-wide CRISPRi-seq screens and generate antibiotic-gene essentiality signatures. Expectedly, genes involved in DNA recombination and repair become more important under fluoroquinolone-induced DNA damage, including *recA*, *recJ*, *recF*, *recO*, *rexAB*, and *ruvAB*. Surprisingly, specific downregulation of the gene encoding the histidine kinase LiaS caused fluoroquinolone hypersensitivity. LiaS is part of the LiaFSR (VraTSR) three-component regulatory system involved in cell envelope homeostasis. We show that LiaS keeps the response regulator LiaR inactive, and that *liaS* deletion causes LiaR hyperphosphorylation and upregulation of the LiaR regulon. We use RNA-seq to refine the LiaR regulon, highlighting the role of heat-shock response and pleiotropic regulator SpxA2 in fluoroquinolone sensitivity. Activating the LiaR-regulon by the cell envelope-targeting antibiotic bacitracin synergized with ciprofloxacin and levofloxacin, restoring sensitivity in fluoroquinolone-resistant strains in vitro. Furthermore, bacitracin/levofloxacin combination therapy is effective in vivo and improved treatment of fluoroquinolone-resistant *S. pneumoniae* infection in a zebrafish meningitis model. These findings offer a starting point for identification and validation of potent combination therapies to treat antibiotic-resistant pneumococcal infections.

The Gram-positive bacterium *Streptococcus pneumoniae* (the pneumococcus) is a clinically significant human pathogen[1]. Pneumococci usually reside asymptomatically in the human nasopharynx, however, under certain conditions can cause disease, particularly in young children, the elderly and the immunocompromised[2]. This includes mild infections such as otitis media and sinusitis, ranging to more severe invasive forms such as pneumonia, meningitis, and bacteremia[2]. *S. pneumoniae* infections remain one of the leading causes of death

Department of Fundamental Microbiology, Faculty of Biology and Medicine, University of Lausanne, Lausanne, Switzerland.
✉e-mail: Jan-Willem.Veening@unil.ch

globally, resulting in approximately half a million deaths per year[3]. Despite available pneumococcal vaccines, which have significantly reduced invasive pneumococcal diseases, antibiotic therapy remains a critical treatment[4].

The first-line treatment strategy for community-acquired pneumonia (CAP) is amoxicillin or doxycycline[5]. If there are comorbidities, treatment extends to a combination with a cephalosporin or macrolide. However, the widespread use of these antibiotics has resulted in the emergence of drug-resistant strains of pneumococci[6,7]. With the increasing prevalence of β-lactam- and macrolide-resistant *S. pneumoniae*, fluoroquinolones have emerged as an important drug class[8]. In cases where there is drug-resistant pneumococcal infection, elderly patients or underlying medical conditions, monotherapy with respiratory fluoroquinolones is indicated for CAP[5]. These synthetic fluoroquinolones also have advantages such as having broad-spectrum antimicrobial activity, good penetration into the respiratory tract and central nervous system, and favorable pharmacokinetic and pharmacodynamic profiles[9].

Fluoroquinolones inhibit DNA synthesis by targeting DNA topoisomerase IV (ParC, ParE) and DNA gyrase (GyrA, GyrB), which are involved in chromosome segregation and DNA supercoiling[10]. In *S. pneumoniae*, the primary target of ciprofloxacin and levofloxacin is topoisomerase IV (ParC subunit), and the primary target for moxifloxacin is DNA gyrase (GyrA subunit)[10]. The formation of DNA-fluoroquinolone-topoisomerase complexes leads to lethal double-stranded breaks (DSBs)[11]. Fluoroquinolone resistance is conferred by spontaneous point mutations in the quinolone resistance-determining regions (QRDRs) of *parC* and *gyrA*[12]. First-step mutations in either *parC* or *gyrA* result in low-level fluoroquinolone resistance, and subsequent second-step mutations in either gene result in high-level resistance[13]. Notably, "first-step" mutants still exhibit susceptibility to fluoroquinolones but carry an elevated risk for the selection of second-step mutations during clinical treatment[13]. Horizontal gene transfer from other streptococcal species, particularly in respiratory samples, can also contribute to resistance[6,7]. The expression of efflux pumps, encoded by *pmrA*, *patA*, and *patB*, has also been reported to be involved in fluoroquinolone resistance[14]. The rates of fluoroquinolone resistance in *S. pneumoniae* currently remain low globally (1–3%), however, higher rates (5–14%) have been noted in certain regions[15].

In 2024, the World Health Organization included *S. pneumoniae* as a priority pathogen for which new antibiotics are needed, with a particular focus on macrolide-resistant pneumococci[16]. The current global antibiotic research and development pipeline faces major challenges such as insufficient funding, which significantly reduces the chances of identifying and developing novel antibiotics. It is therefore important to explore mechanisms that enable the re-use of important antibiotics like fluoroquinolones. Here, we investigated the genome-wide effect of fluoroquinolone stress on the pneumococcus to find such mechanisms. Numerous studies examining the genome-wide effects of antibiotics have shown that antibiotic-induced death is a complex process. Besides the direct drug-target effect, there are additional downstream effects that may disturb many other fundamental aspects of bacterial cell physiology and metabolism and induce various stress responses[17–21]. Similarly, the development of antibiotic resistance, tolerance or changes in drug susceptibility levels may extend beyond direct resistance mechanisms and can be influenced indirectly by other genes and pathways[22,23]. We can therefore examine this antibiotic-specific stress response across the genome to better understand the complex effect of antibiotic pressure and to identify genes that play an important role in bacteria surviving antibiotic stress. Differentially essential genes can be classified as "collateral targets", which are genes that are not direct targets of the antibiotic but globally influence sensitivity to the drug[24]. Collateral targets can be manipulated to boost the potency of current antibiotic interventions. Strategies that can be explored include (1) repurposing existing drugs used to treat other

diseases as co-therapeutics (2) combining current antibiotics with synergistic compounds which boost antibiotic potency, reducing the minimum inhibitory concentration (MIC) or (3) using "antibiotic resistance-breaker compounds" which specifically combat the mechanism of resistance and thus restore the effectiveness of the antibiotic by resensitizing resistant bacteria[22,25,26].

In this study, we used CRISPR interference sequencing (CRISPRi-seq) in *S. pneumoniae* treated with three different fluoroquinolones, to generate a genome-wide profile of gene essentiality under fluoroquinolone-induced stress. Previous RNA-seq and Tn-seq studies failed to identify clear antibiotic stress signatures in *S. pneumoniae*[19]. Part of this surprising lack of coherence in response to antibiotics might be caused by the fact that many important genes are constitutively expressed and essential[19,27,28]. In contrast, CRISPRi can be used to target any gene, including essential genes, and offer a powerful approach to studying drug-gene interactions[20,29,30]. Here, we identified and validated several genes that are not direct targets of fluoroquinolones, but nonetheless influence susceptibility to this class of antibiotics. We show that deletion of genes involved in DNA repair and recombination increases susceptibility to fluoroquinolones, while deleting the translation elongation factor gene *efp* leads to decreased susceptibility. We provide evidence for the role of the LiaFSR operon in sensitizing the pneumococcus to fluoroquinolones, an operon which has previously only been associated with cell membrane stress and cell wall-targeting antibiotics. We demonstrate that the deletion of the gene encoding the histidine kinase LiaS leads to the upregulation of the LiaR regulon, resulting in fluoroquinolone sensitivity. We further extend this finding to design a synergistic antibiotic combination therapy with levofloxacin and bacitracin, a cell membrane-targeting antibiotic, which resensitized clinical levofloxacin-resistant pneumococcal strains both in vitro and in vivo.

## Results

### CRISPRi-seq reveals genes involved in fluoroquinolone susceptibility at a genome-wide level

To identify genes involved in fluoroquinolone-induced stress in *S. pneumoniae* D39V, antibiotic-CRISPRi-seq screens were performed against a panel of three fluoroquinolones: ciprofloxacin (CIP), moxifloxacin (MOX), and levofloxacin (LEV). The CRISPRi system is IPTG-inducible, and the library contains 1498 sgRNAs targeting almost all operons present in the D39V genome[27,29]. Induction of dCas9 results in transcriptional repression of both essential and non-essential genes across the genome. This ultimately results in a pool of 1498 unique hypomorphic mutant strains with a single operon knocked down and allows for the determination of antibiotic-gene interactions at a genome-wide level.

The library was grown in standard C + Y lab medium, in the absence and presence of IPTG and at sub-inhibitory concentrations of each fluoroquinolone (Supplementary Fig. 1). Cultures were harvested at the exponential phase, genomic DNA isolated and single guide RNA (sgRNA) abundance quantified by Illumina Sequencing (Fig. 1A). Statistical analysis of sgRNA counts were performed using 2FAST2Q[31] and DeSeq2[32].

Principal component analysis confirmed successful IPTG induction and showed that antibiotic-treated and untreated samples formed distinct clusters (Supplementary Fig. 2A). Furthermore, a clear pattern of conditional gene essentiality was observed when comparing the genome-wide signatures between fluoroquinolone-treated and non-antibiotic-treated samples (Supplementary Fig. 2B). Scatterplots show fitness scores in the presence and absence of each fluoroquinolone (Supplementary Fig. 2C–E).

We then focused on identifying the differentially essential genes that were shared across all three antibiotic treatments (CIP, MOX, LEV), to find those specifically important for fluoroquinolone-induced stress. In total, 44 sgRNAs (operons) were found to be significantly

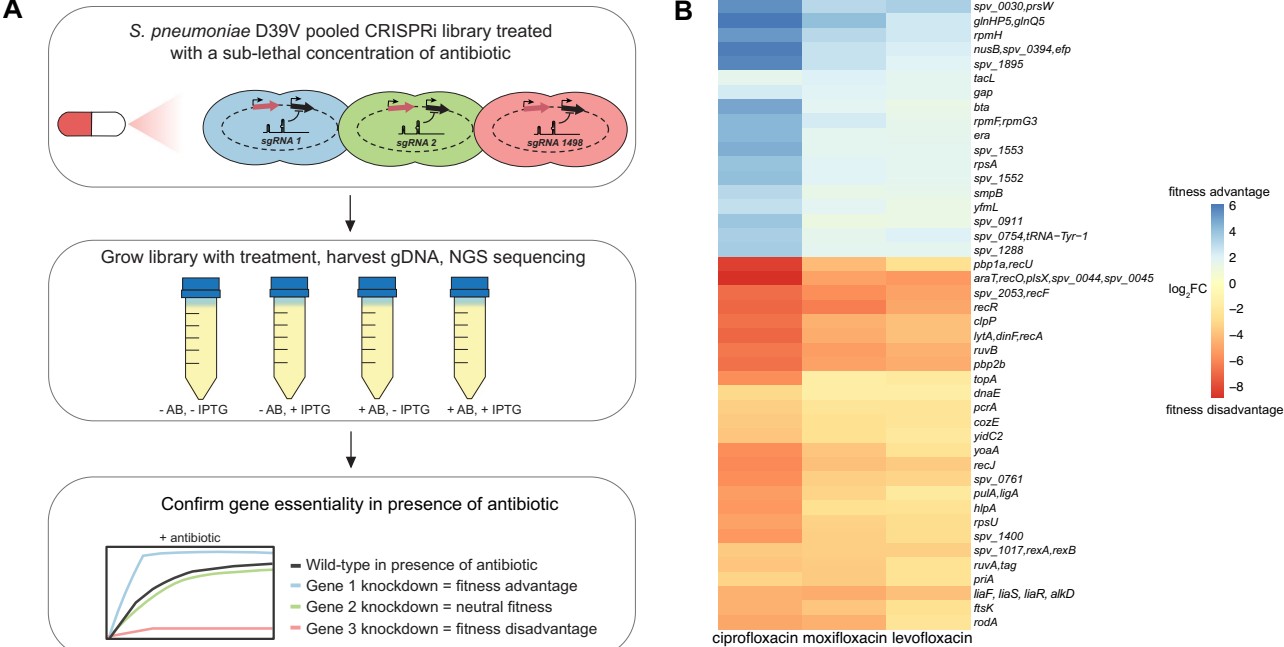

**Fig. 1 | CRISPRi-seq identifies pneumococcal genes influencing fluoroquinolone susceptibility at a genome-wide level. A** Schematic overview of the chemogenomic screen workflow. The *S. pneumoniae* CRISPRi library was treated with a sub-lethal dose of each antibiotic (AB) in the absence or presence of IPTG to induce dCas9. The results of the pooled CRISPRi-seq screen were then validated by testing individual mutants. **B** Heatmap depicting all the sgRNAs (representing operons that have been transcriptionally repressed) that showed a significant differential fitness effect upon antibiotic treatment shared between all three fluoroquinolones as compared to no antibiotic treatment ($|\Delta log_2FC| > 1$, $P_{adj} < 0.05$; DESeq2[32] differential enrichment analysis using a negative binomial generalized linear model with two-tailed P-values adjusted for false discovery rate (FDR)). Source data available in Supplementary Data 1.

differentially essential across the three fluoroquinolones ($|log_2$ fold change ($log_2FC$) $|>1$, $P_{adj} < 0.05$) (Fig. 1B). Of these, 26 operons were found to be depleted (red), indicating that the knockdown of these operons resulted in a relative fitness disadvantage in the presence of fluoroquinolones. Several depleted operons contained genes involved in DNA repair and recombination (e.g., *recA*, *recF*, *recJ*, *ruvAB* and *rexAB*), in line with fluoroquinolones causing DNA damage. Direct fluoroquinolone targets, *parC* and *gyrA* did not appear as differential fitness hits from the screen as these were already highly essential without antibiotic treatment (Supplementary Data 1). The other 18 operons were enriched (blue), suggesting that when knocked down, they resulted in a relative fitness advantage in the presence of fluoroquinolones (Fig. 1B).

Interestingly, the antibiotic stress signature was similar between the three different fluoroquinolones, suggesting that antibiotic-gene interactions are dependent on the mode of action of the antibiotic (Fig. 1B, Supplementary Fig. 2B). The profiles also indicate that antibiotic-induced pressure is not restricted to the direct target of the antibiotic and does not induce a single stress response but is, in fact, a genome-wide effect[17,18,20,30].

Based on the CRISPRi-seq data, 21 operons of some of the top hits were selected for phenotypic validation. We selected genes based on their functional category; some known to be involved in the DNA repair and recombination pathway were selected to validate the screens, while some genes with no known role in this pathway were selected to test for potential involvement in fluoroquinolone stress. Individual CRISPRi knockdown strains were constructed and tested with sub-lethal concentrations of the three fluoroquinolones (Supplementary Fig. 3). The growth phenotypes, which showed either increased or decreased sensitivity to each of the three fluoroquinolones, were confirmed in 20 out of the 21 strains. This further corroborated the reliability of the pooled CRISPRi-seq screens.

## Deletion of *recF* and *recJ* confers fluoroquinolone hypersensitivity in *S. pneumoniae*

The RecFOR system in *S. pneumoniae* plays a key role in repairing single-stranded DNA (ssDNA) gaps and is part of the broader homologous recombination (HR) repair pathway[33–37]. The ssDNA-specific exonuclease, RecJ, works in combination with RecFOR to extend the ssDNA region, enhancing the ability of RecFOR to load RecA[38,39] (Fig. 2A). Strikingly, most genes in the HR pathway were hits in the fluoroquinolone screen. Two of these genes, *recF* and *recJ*, were selected for further validation. Individual gene depletion strains were constructed by expressing *recF* or *recJ* ectopically under the control of an IPTG-inducible $P_{lac}$ promoter, while the respective gene was deleted from its native locus. Both the *recJ* and *recF* depletions showed hypersensitivity to LEV, which was rescued following the addition of IPTG (Fig. 2B, C). This shows that the RecFOR pathway is required for the pneumococcus to survive fluoroquinolone stress, indicating a role of the ssDNA gap repair pathway in DNA break repair following fluoroquinolone treatment.

## Inhibiting translation elongation ameliorates the effect of fluoroquinolone stress

The CRISPRi-seq screens revealed that repression of the operon containing genes *nusB* (transcription termination protein), *efp* (translation elongation factor P), and *spv_0394* (uncharacterized function), conferred a relative fitness advantage in the presence of fluoroquinolone antibiotics (Fig. 1B). Protein synthesis has been suggested to have a role in facilitating fluoroquinolone-mediated cell death, but the mechanism is not well understood[40,41]. However, continuous protein synthesis is necessary in *Escherichia coli* for certain quinolones to be lethal[41,42]. We therefore hypothesized that deleting the gene *efp* would confer the fitness advantage identified in the CRISPRi-seq screen. We thus constructed an *efp*

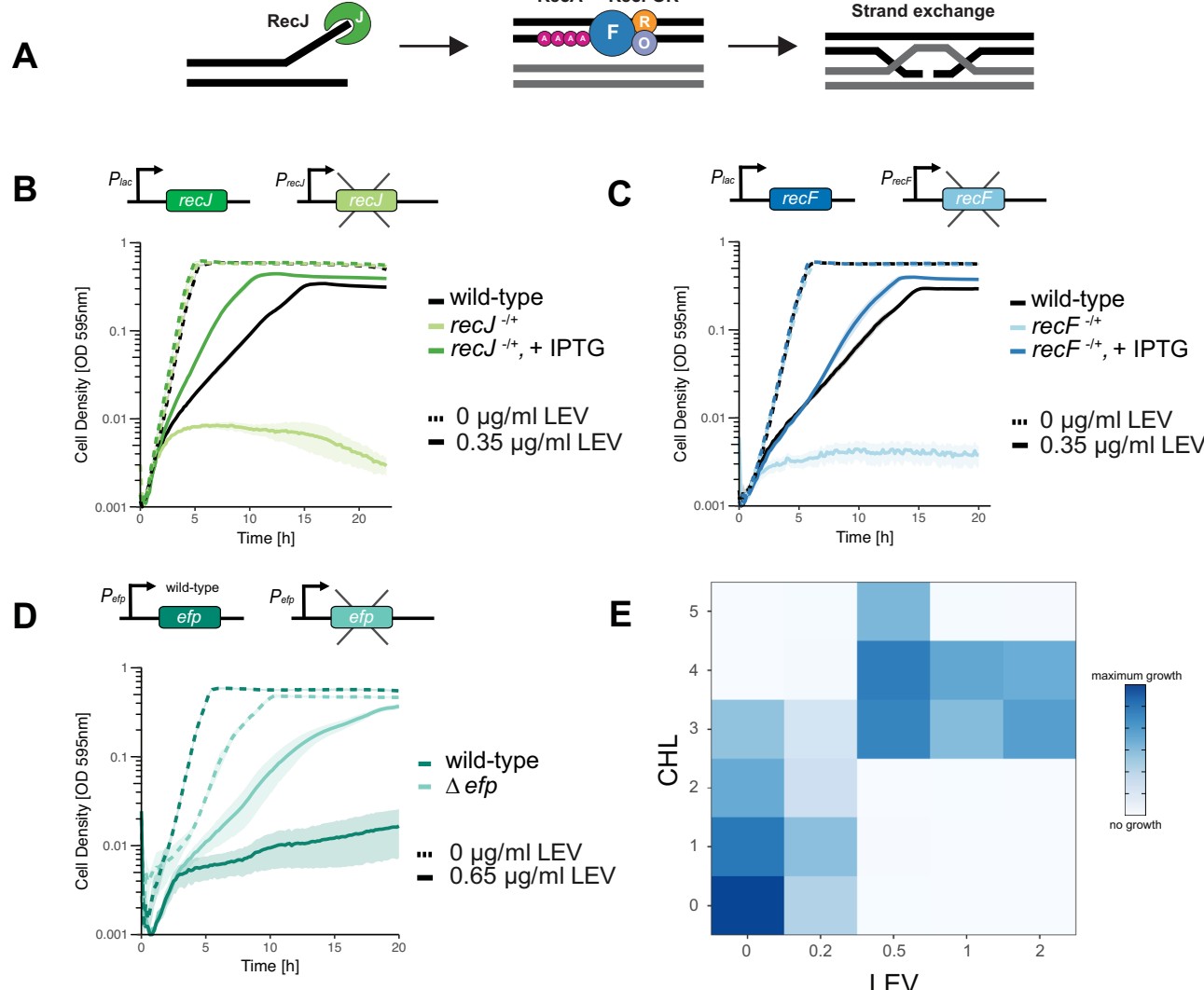

**Fig. 2 | The ssDNA repair complex RecJFOR is crucial for pneumococcal survival during fluoroquinolone treatment. A** At DNA gaps, the ssDNA-specific exonuclease RecJ enlarges the ssDNA region. RecF is a recombination mediator protein and works in complex with RecOR to facilitate the binding of RecA onto the ssDNA. The recombinase RecA, then, promotes homology search, homologous pairing, and strand exchange. Figure adapted from ref. 38. Depletion strains were constructed for (**B**) *recJ* and **C** *recF*, and the mutants were treated with the sub-lethal concentration of LEV used in the screens. Deletion of *recF* or *recJ* resulted in increased sensitivity to LEV, and the phenotype was rescued upon complementation. **D** Deletion of *efp* resulted in a fitness advantage in the presence of a lethal concentration of LEV. Growth curve data represent the mean ± SEM of three biological replicates. **E** Blocking translation elongation using chloramphenicol (CHL) results in an antagonistic interaction with LEV. Source data are provided as a Source Data file.

deletion strain and grew the mutant in the presence of 1×MIC of LEV (0.65 μg/ml) (Fig. 2D). Consistent with our hypothesis, the Δ*efp* mutant survived at higher concentrations of LEV compared to wild-type, confirming the fitness advantage identified in the CRISPRi-seq screens.

The *efp* result suggests that reduced translation elongation may rescue fluoroquinolone-induced damage, suggesting that fluoroquinolones may be antagonized by certain protein synthesis inhibitors. To test this, we examined the drug-drug interaction between LEV and chloramphenicol (CHL). CHL binds to the 50S ribosomal subunit, inhibiting the elongation step in protein translation, and has been shown previously to block the lethality of certain quinolones (Fig. 2E)[41–43]. As shown by checkerboard assays, LEV showed an antagonistic interaction with CHL. These results suggest that the elongation step in protein synthesis may be required for the lethality of certain fluoroquinolones like LEV in *S. pneumoniae*.

## Absence of the histidine kinase LiaS causes hypersensitivity to fluoroquinolones

Interestingly, the screen identified the LiaFSR operon to be important under fluoroquinolone-induced stress (Fig. 1B). The LiaFSR operon, characterized as lipid II-interacting antibiotics sensor and response regulator[44] is a two-component regulatory system in *S. pneumoniae* and plays a key role in responding to cell envelope stress. It is homologous to the LiaFSR system in *Bacillus subtilis*, the VraTSR (vancomycin resistance associated) system in *Staphylococcus aureus*[45] and the CesSR (cell envelope stress) system in *Lactococcus lactis*[46]. The *lia* system is generally induced by cell wall targeting antibiotics, particularly those that disrupt the lipid II cycle, such as bacitracin (BAC), vancomycin and antimicrobial peptides (AMPs)[47]. In the pneumococcus the operon is composed of LiaS, the membrane-localized histidine kinase (HK) and LiaR, the DNA-binding response regulator (RR) (Fig. 3A). A third protein, LiaF, is also involved in signal transduction,

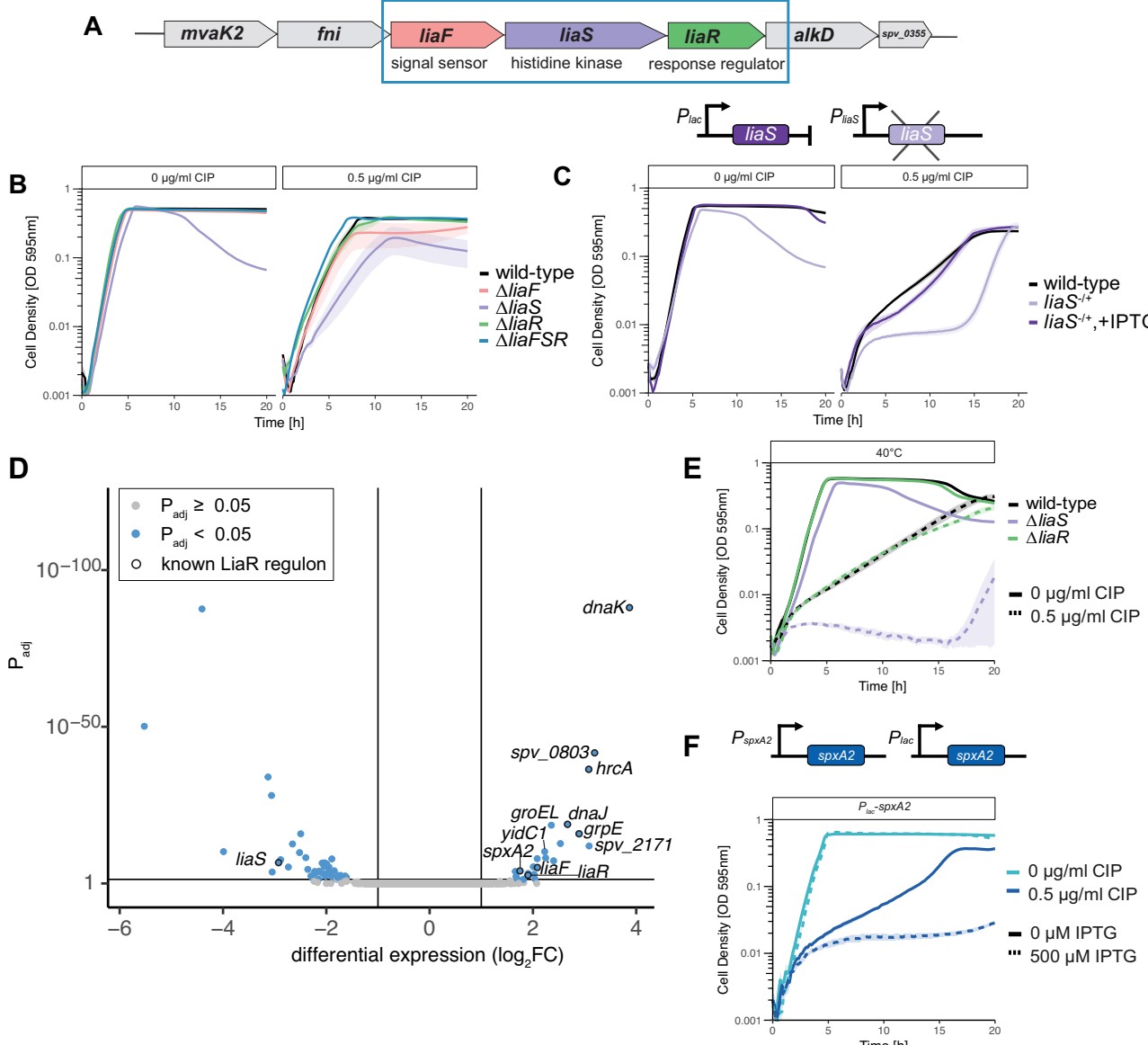

**Fig. 3 | Deleting *liaS* of the LiaFSR three-component sensor system in *S. pneumoniae* increases sensitivity to fluoroquinolone treatment through the upregulation of the LiaR regulon. A** The LiaFSR operon consists of three genes, *liaF* a proposed signal sensor, *liaS* (HK) and *liaR* (RR). **B** Growth curves of deletion mutants Δ*liaF*, Δ*liaR* and Δ*liaFSR* treated with sub-lethal dose of CIP showed sensitivity equal to wild-type (WT). Only the Δ*liaS* mutant showed increased sensitivity to CIP. **C** Complementation of *liaS* restored CIP sensitivity back to WT levels. **D** RNA-seq revealed a set of genes upregulated in a Δ*liaS* mutant, including the entire known LiaR regulon, highlighted by a black outline. Statistical significance

was defined as |log2FC|>1, and $P_{adj}$ < 0.05 (DESeq2[32] differential enrichment analysis using a negative binomial generalized linear model with two-tailed *P* values adjusted for false discovery rate (FDR)). Source data available in Supplementary Data 2. **E** At 40 °C, Δ*liaS* mutants are more susceptible to CIP than WT and the Δ*liaR* mutant. **F** Overexpression of *spxA2* from an IPTG-inducible promoter by addition of IPTG (dashed line) caused increased sensitivity to a sub-lethal concentration of CIP. Growth curve data represent the mean ± SEM of three biological replicates. Source data are provided as a Source Data file.

but its exact role is unclear[48]. It has been suggested that in *B. subtilis*, LiaF may act as a negative modulator of the system[44]. Since the *lia* system has not been previously associated with DNA damage, and as two-component systems are generally not involved in sensing DNA damage[49], we sought to further investigate the role of LiaFSR in fluoroquinolone stress.

When the *lia* operon in an individual CRISPRi mutant strain was transcriptionally repressed, a significant growth defect was observed (Supplementary Fig. 3A). We note that the used sgRNA targets *liaF*, the first gene in the operon. To identify which of the gene(s) within the operon was responsible for fluoroquinolone hypersensitivity, gene replacement mutants were constructed individually for *liaF*, *liaS*, *liaR*, and the full operon (all three genes deleted). No differences in growth

phenotypes were observed for Δ*liaF* or Δ*liaR* mutants with or without sub-lethal concentrations of CIP treatment (Fig. 3B). However, the Δ*liaS* mutant exhibited a slight growth defect in the absence of CIP, which was strongly exacerbated with CIP treatment (Fig. 3B). Interestingly, when the entire operon was deleted (Δ*liaFSR*), the strain grew similar to wild-type, in the presence or absence of antibiotic. This suggests that transcriptional repression of the entire operon by CRISPRi led to relatively larger effects on *liaS* compared to *liaR*.

To confirm that fluoroquinolone hypersensitivity was solely a result of deleting *liaS*, a depletion-complementation strain was constructed by placing *liaS* under an IPTG-inducible $P_{lac}$ promoter at an ectopic locus while *liaS* was deleted from its native locus. The depletion indeed showed a growth defect and hypersensitivity to CIP. Upon

complementation by the addition of IPTG, growth was restored in both the presence and absence of CIP (Fig. 3C).

## Activation of the LiaR regulon causes CIP hypersensitivity

We hypothesized that deletion of *liaS* alters the expression of the LiaR regulon, and this would lead to increased sensitivity to fluoroquinolone antibiotics. To test this, we conducted RNA-seq on both wild-type and Δ*liaS* strains (Fig. 3D). Interestingly, we found that all genes known to be part of the LiaR regulon were upregulated in the *liaS* mutant. Together, this suggests that deletion of *liaS* causes activation of LiaR. In addition, 23 other genes were significantly upregulated ($\log_2FC > 1$, $P_{adj} < 0.05$) and 47 were significantly downregulated ($\log_2FC > -1$, $P_{adj} < 0.05$), indicating the pleiotropic effects of constitutive upregulation of the LiaR regulon.

Since many members of the HrcA heat-shock regulon[50,51] were also upregulated in the Δ*liaS* mutant, we tested whether increasing the growth temperature to 40 °C would also cause increased sensitivity to CIP. Indeed, *S. pneumoniae* grown at 40 °C was more susceptible to CIP than cells grown at 37 °C (cf. Fig. 3C, E). Strikingly, the Δ*liaS* mutant was hypersensitive to CIP at 40 °C (Fig. 3E). This suggests that activation of the heat-shock regulon is not the only contributing factor leading to increased CIP susceptibility of a Δ*liaS* mutant.

To determine if upregulation in the *liaS* mutant of any of the non-heat-shock genes may also contribute to CIP sensitivity, we generated individual gene overexpression strains. The gene was maintained at the native locus, while a second copy of the individual gene was cloned at the ectopic ZIP locus, under the control of the inducible $P_{lac}$ promoter. Strains were cultured in the presence of 500 μM of IPTG to overexpress transcription of the selected genes. Of the 9 genes that were tested, *spxA2* showed the greatest sensitivity to CIP when overexpressed (Fig. 3F, Supplementary Fig. 4). SpxA2 is annotated in *S. pneumoniae* as a non-essential, transcriptional regulator[28]. Spx proteins (suppressor of ClpP and ClpX) are global transcriptional regulators that act by binding RNA polymerase to activate or repress transcription in response to the cell's redox state[52–54]. In *Bacillus spp.*, Spx proteins act as a disulfide/thiol switch for the redox-sensitive transcriptional regulation of genes[55]. *Streptococcus spp.* have two homologs, SpxA1 and A2[53,56]. SpxA1 is reported to be the main regulator of oxidative stress genes[56–58]. SpxA2 appears to primarily function in cell wall and cell membrane homeostasis, while playing a secondary, back-up role in oxidative stress response[56,58]. Together, these results show that LiaR-dependent activation of both the SpxA2 and HcrA regulons contributes to reduced CIP susceptibility.

## The LiaR regulon is activated by bacitracin, a lipid II cycle-inhibiting AMP in *S. pneumoniae*

To monitor the activity of the LiaR regulon in live cells, a reporter strain was constructed in which luciferase (*luc*) was expressed at the ectopic CEP locus under the control of the *spv_0803* promoter, one of the strongest LiaR-dependent promoters (Fig. 4A)[51]. Several compounds known to disrupt the cell envelope were tested to determine which would induce a luciferase signal in the $P_{spv\_0803}$-*luc* strain and thus represent induction of the LiaR regulon. From the compounds we tested, BAC and the AMP LL-37 induced the $P_{spv\_0803}$-*luc* strain (Supplementary Fig. 5). Induction of the luciferase fusion strain, $P_{spv\_0803}$-*luc*, by BAC displayed the strongest luciferase signal (Fig. 4A). The $P_{spv\_0803}$-luc strain did not show increased luminescence with CIP, indicating that LiaR was not induced by fluoroquinolones (Fig. 4A).

## LiaS is regulated by LiaF and maintains LiaR in an unphosphorylated state

RNA-seq data indicated that deletion of *liaS* resulted in upregulation of the LiaR regulon. This implied that LiaS either prevents LiaR phosphorylation by kinase occlusion or acts primarily as a LiaR phosphatase rather than a kinase. This model also implies that in the absence of LiaS, LiaR is phosphorylated by an alternative phosphate donor (Figs. 3D and 5). Indeed, it has been suggested for other Gram-positive bacteria that LiaR can be phosphorylated by acetyl phosphate, which is abundantly present in the bacterial cytoplasm[59–61]. To test expression levels of the LiaR regulon in the absence of the genes of the *liaFSR* operon, deletion mutants of individual genes *liaF*, *liaS*, *liaR*, and the entire operon (*liaFSR*) were constructed by replacing genes with an erythromycin marker, in the background of the $P_{spv0803}$-*luc* reporter strain. The Δ*liaS* mutant showed an upregulation of expression from $P_{spv0803}$-*luc*, similar to BAC treatment. This further supports that LiaR is phosphorylated in the absence of LiaS.

Interestingly, $P_{spv0803}$-*luc* is also slightly activated in the Δ*liaF* mutant and is no longer activated by BAC. This suggests that LiaF is essential for LiaS signal transduction to LiaR (Fig. 4A) and that in the absence of LiaF, a small conformational shift in LiaS takes place, leading to partial release of LiaR from the LiaS/LiaR complex. Predictions of the LiaFSR complex using Alphafold3[62] suggest an altered LiaS/LiaR conformational state when bound to LiaF, suggesting that disruption of the LiaS/LiaF interaction, for instance due to BAC-induced membrane stress, could result in the release of LiaR in the cytoplasm and its subsequent phosphorylation, dimerization and DNA-binding (Fig. 5). While this model is hypothetical, it is in line with a model proposed for the LiaS/R system of *Listeria monocytogenes*[60].

To confirm that constitutive phosphorylation of LiaR upregulates the LiaR regulon, a phosphomimetic *liaR* allele was constructed by an amino acid exchange at position 53, changing the conserved aspartate for a glutamate (LiaR*, D53E)[60]. This allele was placed under the control of a $P_{lac}$ promoter at the ectopic ZIP locus, in the background of the $P_{spv\_0803}$-*luc* reporter strain. Upon induction with IPTG, the phosphomimetic strain displayed increased expression of luciferase from $P_{spv\_0803}$-*luc*, indicating that phosphorylated LiaR indeed upregulates genes of the LiaR regulon (Fig. 4B). In addition, the induced phosphomimetic strain displayed a similar growth phenotype to the Δ*liaS* mutant with lysis beginning around 12 hours of growth (Fig. 4B). To test if phosphorylation of LiaR is synthetic lethal with fluoroquinolone stress, the phosphomimetic strain was treated with sub-lethal concentrations of CIP. The growth defect observed upon induction of LiaR* was greater than in the non-induced strain (Fig. 4C). This showed it is indeed the phosphorylation of LiaR and, therefore, induction of the LiaR regulon that sensitizes pneumococci to CIP stress.

## Activation of the LiaR regulon by bacitracin resulted in synergy with CIP

As the upregulation of the LiaR regulon resulted in increased CIP sensitivity, we hypothesized that by upregulating the regulon with BAC, we could increase sensitivity to CIP. To establish if the drugs would behave synergistically in combination, disc diffusion assays were performed. *S. pneumoniae* D39V was plated on CAB, supplemented with or without BAC [0.5 μg/ml] and with or without a CIP disc [5 μg/ml]. The diameter of the zone of inhibition was compared and a clear difference was observed between the CIP disc only (6 mm) and CIP + BAC combination (12 mm) (Fig. 6A). Several clinical pneumococcal strains of different serotypes and isolated from different sites of infection were tested, and the ability of CIP and BAC to synergize was consistently observed (Supplementary Fig. 6). We also tested for the development of resistance using concentration gradient plates. While LEV and BAC alone, at 2× MIC, rapidly selected for resistant colonies, the combination of the two resulted in no observed resistant colonies (Supplementary Fig. 7). Furthermore, we did not observe a synergy between CIP and BAC in *B. subtilis* and *S. aureus* strains, consistent with the fact that they do not have an SpxA2 homolog

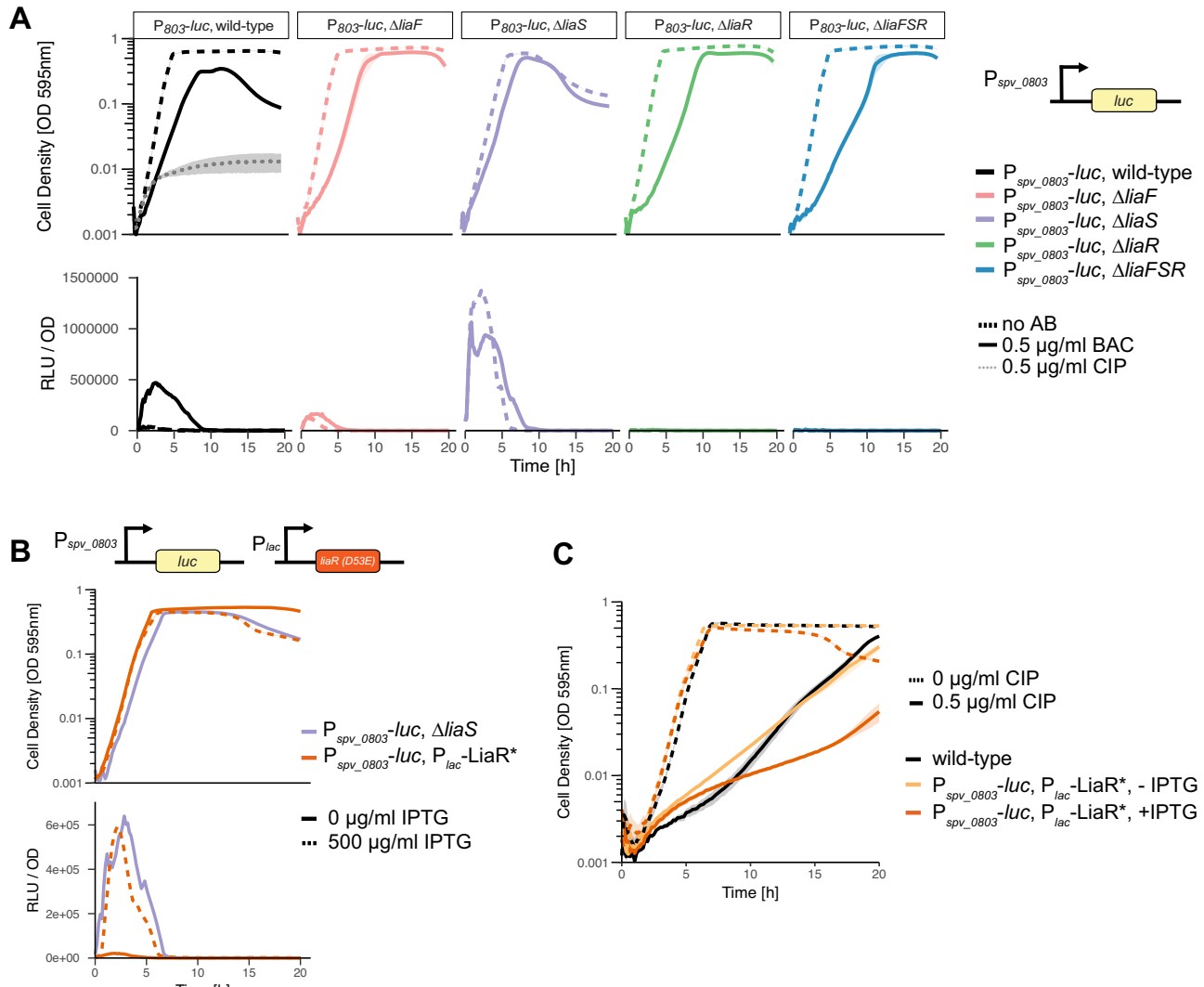

**Fig. 4 | LiaS acts primarily as a phosphatase or inhibitor of LiaR phosphorylation. A** The *luc* reporter gene was placed under the control of the promoter of *spv_0803*, a gene that is strongly induced by LiaR-P. Upon bacitracin (BAC) treatment, luciferase was highly expressed, indicating induction of the LiaR regulon. Deletion of *liaS* results in overexpression of luciferase. A Δ*liaF* mutant displays slight induction of the LiaR regulon, even in the absence of BAC. **B** A phosphomimetic version of LiaR (LiaR*, D53E) was constructed under the control of the IPTG-inducible *Plac* promoter. The strain *Pspv_0803-luc, Plac-liaR** phenocopied a Δ*liaS* mutant in both growth phenotype and induction of the regulon. **C** Activation of the LiaR regulon by expression of LiaR* results in increased sensitivity to ciprofloxacin (CIP). Growth curve data represent the mean ± SEM of three biological replicates. Source data are provided as a Source Data file.

nor the HrcA heat-shock response under control of their LiaR/VraR regulons (Supplementary Fig. 8)[45,63,64].

## Bacitracin synergizes with levofloxacin, rendering resistant strains susceptible

Since LEV is clinically more used for the treatment of pneumococcal pneumonia[5], we performed checkerboard assays with LEV and BAC and calculated fractional inhibitory concentration index (FICI) scores to determine the drug-drug interaction[65]. For wild-type *S. pneumoniae* D39V, the FICI of the combination was 0.5, indicating it was indeed a synergistic interaction (Fig. 6B). We then tested this combination against two highly resistant LEV clinical strains to see if this synergy extends to resistant strains. The checkerboard assays for strains FQR_25191 (sputum isolate, serotype 31) and FQR_40109 (blood isolate, serotype 24 F) have LEV MICs of 16 µg/ml. However, when combined with BAC, the LEV MIC was reduced to 4 µg/ml (Fig. 6C, D). This combination treatment strategy was thus able to increase the sensitivity of LEV-resistant *S. pneumoniae* to LEV in vitro.

## Improved treatment of fluoroquinolone-resistant *S. pneumoniae* with bacitracin-levofloxacin combination therapy

To investigate whether the BAC and LEV combination acted synergistically in vivo, we used a zebrafish embryo pneumococcal meningitis infection model[66]. Zebrafish embryos were infected with *S. pneumoniae* D39V and the LEV-resistant strain FQR_25191. The zebrafish embryos were then treated with LEV or BAC individually, and with the combination of both (Fig. 7A). For D39V at 72 hours post injection (hpi), the percentage survival of zebrafish embryos treated individually with LEV was 33.3% and with BAC, it was 26.7% (Fig. 7B). These survival rates were comparable to the DMSO vehicle control at 15.6% survival. However, the combination of BAC and LEV increased survival to 75.6%. For the fluoroquinolone-resistant strain FQR_2519, we observed that at 72hpi, the percentage survival of zebrafish embryos treated individually with LEV was 42.2% and with BAC, it was 35.6% (Fig. 7C). These survival rates were comparable to the DMSO vehicle control at 37.8% survival. Strain FQR_25191 also showed a high BAC MIC, which was 16 µg/ml compared to 8 µg/ml in D39V (Fig. 6B, C), which explains why

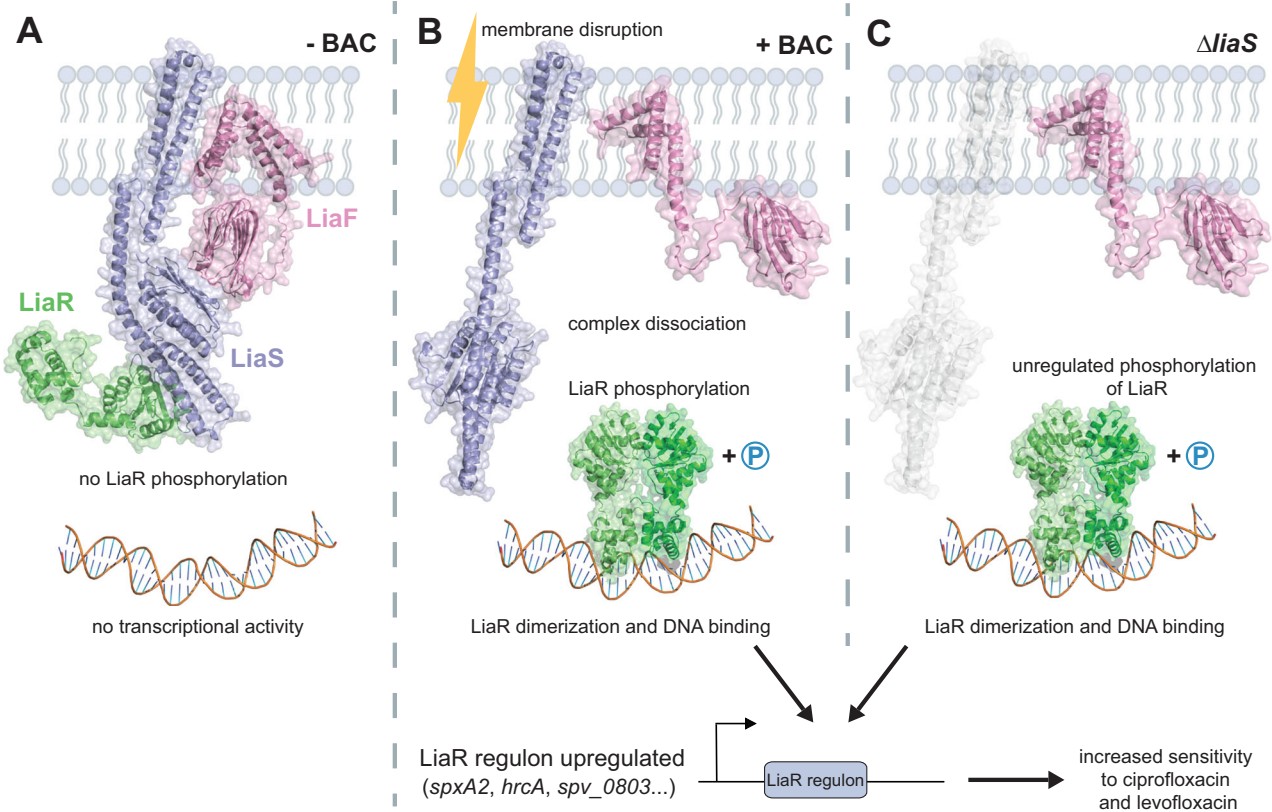

**Fig. 5 | Schematic representation of the role of the LiaFSR three-component regulatory system in fluoroquinolone susceptibility. Hypothetical model of the LiaFSR system modeled with AlphaFold. A** In the absence of a stimulus, LiaFSR remains an intact complex. LiaS exhibits phosphatase activity, ensuring LiaR remains unphosphorylated and the regulon remains inactive. LiaF senses cell envelope stress, resulting in conformational changes in LiaS. **B** Upon cell membrane disruption, for example by bacitracin, the complex dissociates. LiaS no longer dephosphorylates LiaR, allowing LiaR to become phosphorylated, dimerize and bind to DNA to activate transcription of the genes within the LiaR regulon. **C** In the absence of LiaS (Δ*liaS*), there is constitutive phosphorylation of LiaR, likely due to the metabolite acetyl phosphate present in the cell. The LiaR regulon is highly upregulated, and the overexpression of these genes, specifically heat-shock genes and *spxA2*, result in increased sensitivity to ciprofloxacin and levofloxacin.

BAC treatment did not improve survival in this strain. However, the combination of BAC and LEV increased survival to 57.8%. Survival was directly correlated with a reduction in CFU burden, and the synergistic effect of LEV and BAC can be seen in lower bacterial burden compared to treatment with each antibiotic alone (Supplementary Fig. 9). Together these results indicate that the LEV and BAC combination also acts synergistically in vivo, and that BAC can potentiate LEV to more effectively kill fluoroquinolone-resistant pneumococci.

## Discussion

A comprehensive understanding of the genome-wide factors that modulate antibiotic susceptibility is essential to inform the development of new treatments or strategies for pneumococcal infections. To address this, we utilized CRISPRi-seq to determine the gene fitness of *S. pneumoniae* in the presence of fluoroquinolone antibiotics. We show that the biological insight derived from these antibiotic-gene interactions can potentially be extended to the clinical setting and create potent drug combinations.

Our results show that genes involved in DNA recombination and repair, such as *recF* and *recJ*, become conditionally essential under fluoroquinolone-induced DNA damage (Fig. 2). We also show that deletion of *efp* plays a role in ameliorating fluoroquinolone-induced stress (Fig. 2). This corresponds to a previous study that demonstrated deleting the gene *ssrA*, which recycles stalled ribosomes like *efp*, allowed *S. pneumoniae* to survive fluoroquinolone-mediated killing[40].

A surprising finding in this study is the implication of the LiaFSR three-component system in fluoroquinolone susceptibility. By upregulating the LiaR regulon, either through deletion of the gene encoding the histidine kinase LiaS or by overproducing a phosphomimetic variant of LiaR, hypersensitivity to CIP was observed (Figs. 3 and 4). Based on this data, we propose a speculative model in which LiaR is kept inactive by LiaS, either by kinase occlusion or by acting primarily as a phosphatase (Fig. 4). Upon the detection of membrane stress by LiaF, LiaS changes conformation, making it unable to further dephosphorylate LiaR. LiaR then becomes phosphorylated and activates transcription of its regulon. This model makes sense physiologically for a membrane homeostasis sensing system, as upon membrane damage, the system does not require an intact phosphotransfer reaction between membrane-based kinase and cytoplasmic response regulator, which might be compromised under such conditions (Fig. 5). Similar models have been proposed for *Listeria monocytogenes* and *B. subtilis*[44,59,67].

Previous experimental identification of the pneumococcal LiaR regulon utilized DNA microarrays and involved comparing wild-type with a *liaR* mutant[51]. Here, RNA-seq was used under LiaR hyperactive conditions (in a Δ*liaS* mutant), providing a more complete picture of the LiaR regulon (Supplementary Data 2). We investigated genes that were upregulated in the RNA-seq data by generating individual gene overexpression strains. We found that overexpression of the transcriptional regulator *spxA2* was a key contributor to the increased CIP or LEV sensitivity (Fig. 3). It is tempting to speculate that cell

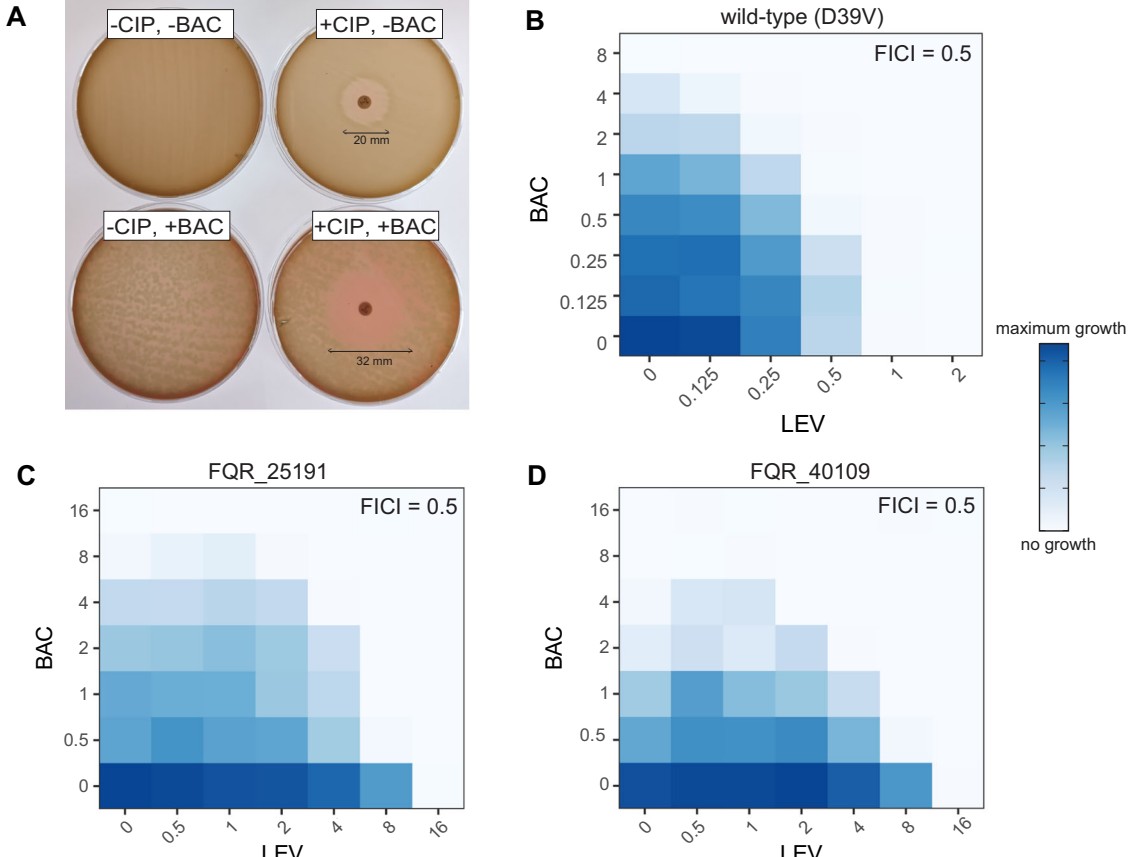

**Fig. 6 | A combination of bacitracin together with ciprofloxacin or levofloxacin increases the potency of these fluoroquinolones. A** Ciprofloxacin (5 mg/ml) disc diffusion assays show an increased zone-of-inhibition of strain D39V with the addition of 0.5 µg/ml bacitracin (BAC). Checkerboard assays with levofloxacin (LEV) and BAC indicate a synergistic interaction (FICI = 0.5) for (**B**) D39V and the fluoroquinolone-resistant strains (**C**) FQR_25191 and **D** FQR_40109 (see Methods).

membrane remodeling (Supplementary Fig. 10) induced by activation of the LiaR regulon[68] leads to better penetrance of certain fluoroquinolone antibiotics such as CIP and LEV thereby causing increased susceptibility. The exact role of SpxA2 in this requires further investigation.

The insights into the regulation of the LiaFSR regulon established in this study allowed the rational design of a synergistic combination therapy. BAC combined with either LEV or CIP, resensitized a panel of fluoroquinolone-resistant clinical *S. pneumoniae* strains, of different serotypes and clinical origin (Fig. 6 & Supplementary Fig. 6). We investigated the interaction between fluoroquinolones and bacitracin in combination with other drugs, and the synergy appeared to be specific to the bacitracin and fluoroquinolone combination (Supplementary Fig. 11). Interestingly, in a systematic drug combination screen, BAC was also found to have synergy with CIP in *S. pneumoniae* D39V[69]. Here we show that this combination strategy also worked in vivo in a zebrafish meningitis model, increasing the potential for its clinical relevance (Fig. 7). Combining BAC with LEV could reduce the necessary administered doses required, potentially minimizing side effects and making systemic use of BAC clinically relevant. Additionally, investigations into the design of BAC analogs with improved systemic safety are also currently underway[70]. LiaS inhibitors could further enhance the effectiveness of fluoroquinolones in *S. pneumoniae*. It has already been demonstrated in *S. aureus* that a kinase inhibitor can be used to inhibit LiaS and this potentiated the effect of several cell wall targeting antibiotics[71–73]. It is interesting to note that the AMP LL-37, part of the human innate immune system, also stimulates the LiaR regulon[74] (Fig. S5). Thus, part of the success of the use of

fluoroquinolones in practice, and the relatively low levels of global resistance currently observed, could be because during infection, the host already triggers the LiaR response, thereby sensitizing bacteria for the antibiotic and preventing rapid selection of resistant mutants (Fig. S7). Using this information, high-throughput screening for new compounds or AMPs that activate the LiaFSR system could lead to promising adjunctive therapies. Once safer alternatives to bacitracin have been identified, conducting studies of clinical relevance of synergy in a murine model would be interesting to explore. Resistant colonies could also not be selected for with the combination of the BAC and LEV (Supplementary Fig. 7,) further making this combination beneficial as it minimizes the development of resistance.

Together, this work showcases that genome-wide CRISPRi screens combined with antibiotic perturbations can give valuable new insights into antibiotic susceptibility and can identify new leads for synergistic combination therapy.

## Methods
### Ethics statement
The research presented here complies with all relevant ethical regulations and protocols as approved by UNIL and complies with the Swiss regulations on animal experimentation (Animal Welfare Act SR 455 and Animal Welfare Ordinance SR 455.1).

### Bacterial strains and culture conditions
All strains and primers used are listed in Supplementary Data 3 and 4, respectively. All pneumococcal strains used in this study are a derivative of the serotype 2 strain *S. pneumoniae* D39V[28], unless stated

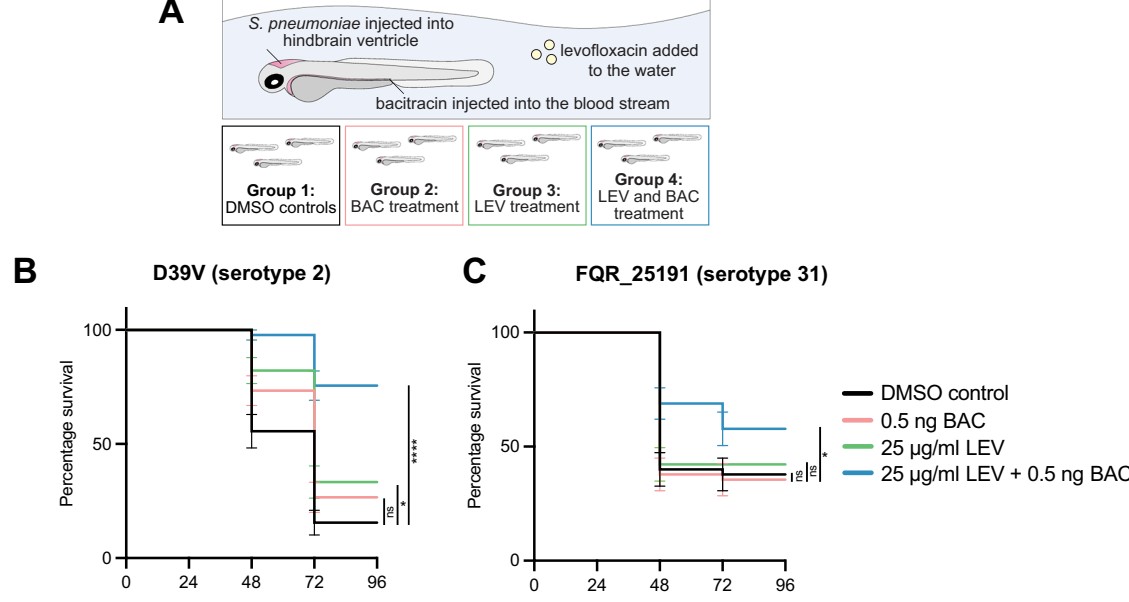

**Fig. 7 | Levofloxacin and bacitracin act synergistically in vivo in a zebrafish embryo meningitis model. A** Experimental set up. *S. pneumoniae* was injected into the hindbrain ventricle of zebrafish larvae at 2 days post-fertilization. Zebrafish were treated at 1 hour post injection with: DMSO vehicle controls (group 1), BAC injected into the bloodstream and DMSO vehicle control added to the water (group 2), LEV added into the water and DMSO vehicle control injection (group 3), or a combination of both LEV added to the water and BAC injection (group 4). **B** Survival curves of zebrafish embryos injected with -300 CFUs of *S. pneumoniae* D39V. Zebrafish embryos were treated with DMSO (control), LEV, BAC or a combination of both BAC and LEV. The data represent the mean ± SEM of three biological replicates with 15 larvae per group ($n = 45$ in total/group); ns = not significant, *$p = 0.0064$, ****$p < 0.0001$; determined by Log-rank test. **C** Survival curves of zebrafish embryos injected with -300 CFUs of fluoroquinolone-resistant *S. pneumoniae* strain FQR_25191. Zebrafish embryos were treated with DMSO (control), LEV, BAC or a combination of both BAC and LEV. The data represent the mean ± SEM of three biological replicates with 15 larvae per set ($n = 45$ in total/group); ns not significant, *$p = 0.0437$; determined by Log-rank test.

otherwise. Strains were cultured in liquid C + Y medium[75] from starting optical density (OD$_{595}$) of 0.01 and incubated at 37 °C until the required OD.

Strains were stocked at OD595 = 0.3 in C + Y with 15% glycerol at −80 °C.

Induction of the isopropyl β-D-1-thiogalactopyranoside (IPTG)-inducible promoter ($P_{lac}$) was conducted by supplementing the medium with IPTG (Sigma-Aldrich) at the required concentration (10 μM–500 μM). For depletion experiments, depletion strains were grown without inducer until OD$_{595}$ = 0.1 and then diluted 100-fold in fresh media with inducer.

Transformation of *S. pneumoniae* was performed by growing cells in C + Y to OD$_{595}$ = 0.1, adding 1 ng/ml CSP-1 and incubating at 37 °C for 12 min to induce competence. Competent cells (100 μl) were mixed with transforming DNA, incubated at 30 °C for 20 min for DNA uptake, followed by the addition of 900 μl of fresh C + Y medium and recovery at 37 °C for 1.5 hours. Transformants were selected by plating inside Columbia agar supplemented with 3% defibrinated sheep blood (CBA, Thermo Scientific) and the appropriate antibiotic for selection and incubated at 37 °C in 5% CO$_2$ overnight. Antibiotics used for selection were: erythromycin (0.5 μg/ml), spectinomycin (100 μg/ml), and CHL (4 μg/ml). For induction of the IPTG-inducible $P_{lac}$ promoter in some mutants, the media was supplemented with 500 μM of IPTG. Successful transformants were confirmed by colony PCR and Sanger sequencing (Microsynth).

**Strain construction**
**PCR and cloning.** Primers were designed using Benchling Biology Software. PCR reactions contained a final concentration of 1× Phanta Master Mix polymerase (Vazyme), forward primer (0.4 μM), reverse primer (0.4 μM), and template DNA made up to volume with sterile water. Thermal cycling conditions included a 58 °C annealing step and

1 min/kb extension time. Agarose gel electrophoresis was used to resolve DNA samples. If required, PCR product fragments were excised from gels and purified with a PCR clean-up kit (Macherey-Nagel Nucleospin).

**Mutant generation by golden gate cloning.** *S. pneumoniae* mutants were constructed by integrating into the chromosome by double HR using linear DNA fragments containing -1 kb region of homology up and downstream. Linear constructs were generated using Golden Gate assembly. Primers designed for a Golden Gate assembly used the restriction enzymes Esp3I, BsaI, or AarI. The fragments were ligated together using Vazyme T4 DNA ligase. Primers used are listed in Supplementary Data 4. Assemblies were transformed directly into the competent host strain.

**IPTG-inducible strains.** The gene of interest together with its ribosomal binding site were amplified by PCR. The upstream fragment was amplified from a strain integrated with the pPEPZ-$P_{lac}$ plasmid[76], so the upstream region contained $P_{lac}$, and a spectinomycin resistance selection marker, and both up and downstream contained homology to the ZIP locus (ectopic integration locus in the genome causing disruption of the non-coding gene *spv_2417*). All three fragments were assembled using Golden Gate and transformed into a D39V *lacI* strain.

**Deletion strains.** For the deletion of a gene of interest, a deletion assembly consisted of 1 kb upstream and 1 kb downstream of the gene and an antibiotic resistance marker (erythromycin) which replaced the gene.

**Phosphomimetic strain.** The point mutation was generated via overlap PCR using primers OVL10032 and OVL10033.

**Luciferase fusion strains.** To generate the luciferase reporter strain, the 300 bp region upstream of *spv_0803* was amplified by PCR and fused to the *luc* gene encoding luciferase via Golden Gate assembly. The resulting construct was integrated into the ectopic CEP locus.

**Individual CRISPRi mutant strains–sgRNA cloning.** To confirm selected hits, individual CRISPRi mutant strains were generated. First, individual sgRNAs were sub-cloned into the *E. coli* vector pVL3991 by digesting the vector with the restriction enzyme Esp3I and ligating the sgRNA with T4 DNA ligase. The ligation was transformed into *E. coli* and white colonies were selected, as the successful cloning of the sgRNA replaced the mCherry gene. Plasmid DNA was isolated using the Promega PureYield Plasmid Miniprep kit, and after confirmation by sequencing, transformed into the same *S. pneumoniae* background strain as the pooled library.

## Microtiter plate-based growth assays

For *S. pneumoniae* growth assays, cells were precultured in C + Y medium until $OD_{595} = 0.1$ and then diluted 100-fold into 96-well microtiter plates containing fresh C + Y supplemented with appropriate concentrations of antibiotic or inducer as per each experiment. $OD_{595}$ was measured every 10 min at 37 °C for 20 hours using the TECAN Infinite F200/M200Pro plate readers. In the case of luminescence assays, luciferin was added to the media at a concentration of 0.1 mg/mL to monitor gene activation by means of luciferase activity. Luminescence (relative luminescence units [RLU]) and optical density ($OD_{595nm}$) were measured every 10 minutes. Each growth assay was performed in biological and technical triplicate. For growth curves, the average $OD_{595nm}$ value from triplicate samples was plotted. Growth curves were plotted with BactExtract[77] using default parameters.

## Checkerboard analyzes and FIC index determination

Standard two-dimensional checkerboard assays were conducted to determine antibiotic interactions. Antibiotics were serially diluted 2-fold with antibiotic A along the *X* axis and antibiotic B along the Y-axis in a 96-well microtiter plate. Control wells included no antibiotic for zero growth inhibition an antibiotic concentration for maximum inhibition. The inoculation and growth measurement by plate reader was as previously described and the AUC plotted with BactExtract[77] using default parameters.

The FICI[65] was calculated with the following formula: $FICI = (MIC_{AB}/MIC_A) + (MIC_{BA}/MIC_B)$, where $MIC_{AB}$ is the MIC of drug A tested in combination, $MIC_A$ is the MIC of drug A tested alone, $MIC_{BA}$ is the MIC of drug B tested in combination, and $MIC_B$ is the MIC of drug B tested alone. Synergy was defined as a $FICI \leq 0.5$, additive as $0.5 < FICI \leq 1$, indifference as $1 < FICI < 4$ and antagonism as a $FICI \geq 4$. FICI was determined from the average of three biological triplicates.

## Antibiotic essentiality screens for CRISPRi-seq

**Growth of library in test conditions.** A frozen stock of the pooled D39V CRISPRi library previously generated in our lab[27] was diluted 100-fold in fresh C + Y media in 4 biological replicates and grown until $OD_{595} = 0.1$. This pre-culture was then diluted 10-fold in C + Y with and without IPTG (40 μM) to an $OD_{595} = 0.1$. The induced and non-induced cultures were each diluted 150-fold into 15 ml of C + Y in a 15 ml Falcon tube, under four conditions: i) antibiotic plus IPTG, ii) antibiotic minus IPTG, iii) no antibiotic plus IPTG, iv) no antibiotic minus IPTG. The three fluoroquinolone antibiotics used in the screens were MOX, LEV, and CIP at sub-lethal concentrations of 0.06, 0.35, and 0.5 μg/ml, respectively. Sub-lethal concentrations were determined from the growth plots and defined as a concentration below the MIC that resulted in a distinct growth defect compared to no antibiotic treatment. Cultures were grown to $OD_{595} = 0.4$, (-11 generations) and then centrifuged (4000 × *g*, 10 min, 4 °C) to pellet cells. The supernatant

was removed, the bacterial pellet washed once with PBS and stored at −80 °C for genomic DNA (gDNA) extraction.

**Genomic DNA isolation.** The frozen cell pellet stored at −80 °C was resuspended in 50 μl of water, and half the volume was taken for DNA extraction. The aliquot was resuspended in 800 μl of Nuclei Lysis solution (Promega) supplemented with 0.05% SDS, 0.025% deoxycholate (DOC), and 200 μg/mL RNase A. This was incubated at 37 °C for 20 minutes, then 80 °C for 5 minutes to lyse cells and 37 °C for 10 minutes. 250 μl of Protein Precipitation Solution (Promega) was added to the lysate, vortexed vigorously, then incubated on ice for 10 min. Samples were centrifuged (14,000 × *g*, 10 min, 4 °C) to pellet the precipitated protein. The supernatant was transferred into 600 μl isopropanol to precipitate the DNA, which was then collected by centrifugation (14,000 × *g*, 10 min). The DNA pellet was washed once in 70% ethanol, air-dried, and resuspended in molecular-grade water. DNA was stored at −20 °C.

**Sequencing of sgRNAs and data analysis.** These steps were conducted according to the protocol recently published by de Bakker et al. (2022)[29]. Briefly, the sgRNAs were amplified by 12 cycles of PCR, using primers designed to bind to the Illumina read 1 and read 2 elements, which are on either side of the sgRNA. Each primer had Illumina barcodes included, and unique pairs were used for each sample to allow for de-multiplexing after the sequencing run. Library pooling and denaturation were conducted following the Illumina guidelines. The library was sequenced using a high-output 300-cycle kit on an Illumina Miniseq sequencer. The sgRNA abundance was quantified using 2FAST2Q[31] and further statistical analysis of count data was performed using the R-package DESeq2[32], implementing a negative binomial generalized linear model to test for (differences in) $log_2FC$ estimates greater than 1, yielding two-tailed *P* values adjusted for false discovery rate (FDR). For heatmap visualizations, $log_2FC$ values were scaled but not centered to allow for direct inter-condition gene-wise fitness comparisons, while retaining the meaning of positive (fitness gain), negative (fitness loss) and zero (neutral genes) values using the built-in R function scale() prior to plotting. R version 4.0.3 was used.

## CIP disc assays

*S. pneumoniae* strains were grown in C + Y broth to $OD_{595} = 0.1$, and a sterile cotton swab was dipped into the liquid and streaked fully onto CAB plates (Columbia agar supplemented with 3% defibrinated sheep blood). This streaking was repeated four more times, rotating the plate each time to ensure an equal distribution of inoculum and to create an even bacterial lawn. In the cases where BAC was required, 0.5 μg/ml was directly supplemented into the agar before pouring. CIP disks (5 μg/ml, Oxoid) were dispensed onto the surface of the inoculated agar plate and gently pressed down. Plates were incubated for 18 hours at 37 °C and 5% $CO_2$ after which the zone of inhibition was measured. A single colony of *S. aureus* was inoculated into TSB and grown until $OD_{595} = 0.1$–0.2. The culture was streaked onto TSA for the disc diffusion assay. A single colony of *B. subtilis* was inoculated into LB and grown until $OD_{595} = 0.4$. The culture was streaked onto an LA for the disc diffusion assay. Plates with *S. aureus* and *B. subtilis* were incubated at 37 °C overnight, and zone of inhibition measured.

## RNA-seq

Wild-type (D39V) and the Δ*liaS* mutant were grown in C + Y medium to $OD_{595} = 0.3$ at 37 °C in quadruplicate. In total, 2 ml of culture was grown, but 1 ml was used for the extraction. Cultures were centrifuged (4 °C, 10,000 × *g*, 5 min), supernatant removed, and cells snap frozen in liquid nitrogen and stored at −80 °C. Total RNA was isolated using the High Pure RNA Isolation kit (Roche) with minor modifications. Frozen cell pellets were resuspended in Tris-EDTA buffer (400 μl) and then added to tubes containing 10% SDS (50 μl), phenol-$CHCl_3$ (500 μl)

and glass beads. Cells were lysed with a bead beater ($3 \times 45$ sec, with 45 sec pauses) and pelleted by centrifugation (4 °C, $21,000 \times g$, 15 min). The aqueous phase (300 μl) was removed and added to 1.2 ml of lysis/binding buffer and vortexed. 600 μl of the samples were loaded onto columns and centrifuged ($8000 \times g$, 30 s). 100 μl DNase mix (90 μl buffer, 10 μl DNase I) was loaded onto the column filter and incubated for 1 hour at room temperature. Samples were washed once with wash buffer I (500 μl) and twice with wash buffer II (first with 500 μl, then 200 μl) with final centrifugation ($8000 \times g$, 2 min). Samples were eluted with 50 μl elution buffer into a clean tube and incubated for 10 min at room temperature. The quantity and quality of the extracted total RNA were checked by Nanodrop and Fraction Analyzer (Agilent Technologies) and stored at −80 °C.

Four RNA samples of D39V and three samples of the ΔliaS mutant (due to DNA contamination in one sample) were sequenced at the Genomic Technologies Facility, University of Lausanne. rRNA depletion was not conducted on these samples. Stranded RNA-seq libraries were prepared using the Illumina Stranded mRNA Prep reagents (Illumina) with a unique dual indexing strategy and following the official protocol, automated. Sequencing was performed on an Illumina NovaSeq 6000 for 100 cycles, single read. Sequencing data were demultiplexed using the bcl2fastq2 Conversion Software (version 2.20, Illumina). Read quality was checked with FastQC (v0.11.9) and MultiQC (v1.15)[78] before and after trimming off leading and trailing bases below a phred score of 33, cutting regions if average phred scores went below 20 in a sliding window of 5 bases and only keeping reads with a minimum length of 50 bases using Trimmomatic (v0.36)[79]. Reads were aligned to the *S. pneumoniae* D39V reference genome (CP027540) using bowtie2 (v2.4.5)[80], using soft-clipping (with the "--local" option) to account for any remaining partial adapters. Transcript counts were extracted with featureCounts (v2.0.6)[81] and downstream analyses were performed with DESeq2[32] in R (v4.0.3), testing against an absolute $\log_2 FC > 1$ at an alpha of 0.05.

## Protein complex structure predictions and visualization
LiaFSR proteins and complexes thereof were predicted using the AlphaFold3 algorithm[62] running in the AlphaFold Server environment[82]. Proteins were predicted either as singular entities or as multi protein complex as indicated. Visualizations were created using PyMol.

## Concentration gradient plates to select resistant colonies
CAB agar with 3% blood was supplemented with 2×MIC concentrations of either LEV (2 μg/ml), BAC (16 μg/ml) or the combination of LEV and BAC. In a large square petri dish, the first layer of CAB was poured at an angle. Once the first antibiotic-containing layer dried, a second layer of CAB containing no antibiotic was added, resulting in a diffusion gradient. *S. pneumoniae* D39V grown in C + Y until $OD_{595} = 0.3$ was then plated, and glass beads were used to evenly spread the culture on the CAB. Plates were incubated at 37 °C in 5% $CO_2$ overnight.

## Zebrafish husbandry and zebrafish embryos infection experiment
Adult $tra^{b6/b6};nac^{w2/w2}$ (TraNac) zebrafish (*Danio rerio*) were maintained and raised under standard conditions. Transparent TraNac embryos were obtained by natural mating and were collected within the first hours post fertilization (hpf) and kept at 28 °C in E3 medium (5.0 mM NaCl, 0.17 mM KCl, 0.33 mM CaCl·2H2O, 0.33 mM MgCl2·7H2O) supplemented with 0.3 mg/L methylene blue. Sex was not considered in the study design because sex differentiation is not yet complete in zebrafish embryos at 2 days post fertilization (dpf.) Before injection, embryos at 2dpf were mechanically dechorionated if necessary and anaesthetized in 0.02 % (w/v) buffered 3-aminobenzoic acid methyl ester (pH 7.0) (Tricaine; Sigma-Aldrich, A5040). The zebrafish embryos were individually infected by microinjection with 1 nl of *S. pneumoniae*

D39V or *S. pneumoniae* FQR_25191 into the hindbrain ventricle as described previously[66]. Before injection, bacteria were suspended in sterile 0.25 % (w/v) amaranth solution (Sigma-Aldrich; A1016) to aid visualization of the injection process. The number of colony-forming units (CFU) per injection was determined by quantitative plating of the injection volume. After infection, zebrafish embryos were kept in six-well plates at 28 °C with 15 individually injected embryos in each group per well. At 1 hour post injection, infected zebrafish embryos were treated with LEV or DMSO vehicle control added to the water and injection with 1 nl bacitracin or DMSO vehicle into the bloodstream. The mortality rate was determined by monitoring live and dead embryos at fixed time points between 24 and 72 hours post injection (hpi). All experiments were performed in triplicates. All procedures related to zebrafish studies at UNIL comply with the Swiss regulations on animal experimentation (Animal Welfare Act SR 455 and Animal Welfare Ordinance SR 455.1). Survival graphs were generated with GraphPad Prism and analyzed with the log-rank (Mantel-Cox) test. Results were considered significant at $P$ values < 0.05.

To determine the bacterial burden, zebrafish larvae at 2 days post-fertilization were infected with *S. pneumoniae* D39V ( ~ 400CFU) and treated with antibiotic as described before. Larvae were anesthetized in 0.02% Tricaine in egg water, transferred to a 1.5 mL screwcap tube (1 larva per tube) filled with 1.0 mm glass beads (Carl-Roth, Art Nr. A554.1) to 25% capacity of the tubes' volume, placed in a microvial rack, and vigorously shaken (3 times 10 s, 10 s interval) in a bead beater to disrupt the cells and tissues. Subsequently, serial dilution plating was performed on Columbia Blood plates supplemented with 5% defibrinated sheep blood and 10 mg/L colistin sulfate and 5 mg/L oxolinic acid (Oxoid, Cat# SR0126) to inhibit growth of commensal bacteria in zebrafish. The plates were incubated overnight at 37 °C, and CFUs were quantified the following day. All experiments were performed in duplicate. CFU plots were generated with GraphPad Prism and analyzed with an unpaired $t$ test.

## DiI-C12 staining and microscopy
For staining of *S. pneumoniae* strains with DiI-C12[83,84], the protocol was adapted from Wenzel et al. (2018)[85]. Frozen stock strains were diluted 100-fold into fresh C + Y media supplemented with 1 μg/ml DiI-C12, appropriate antibiotic or IPTG concentration required for the strain and grown at 37 °C until $OD_{595} = 0.2-0.3$. The stained cells were washed four times in C + Y supplemented with 1% DMSO and resuspended in 100 μl of this medium. Cells (0.8 μl) were then spotted onto pure agarose (1.2% in PBS) pads and sealed with a glass cover slip. Microscopy acquisition was performed using a Leica DMI8 microscope with a ×100/1.40 oil-immersion objective and a sCMOS camera (Leica-DFC9000GT-VSC08519). Phase contrast images were acquired using transmission light (100 ms exposure). Fluorescent images for the DiI-C12 stained cells were acquired using 100 ms exposure and by excitation with 550 nm laser and the TRITC emission channel (595/40 nm). Images were processed using LasX v.3.4.2.18368 (Leica) and using FIJI v.1.52q (fiji.sc).

## Statistics and reproducibility
Data analysis was performed using R (R-package DESeq2[32]), Prism (Graphpad) and BactExtract[77]. Data shown are represented as mean of at least three biological replicates ± SEM. No statistical method was used to predetermine sample size. For RNA-seq experiments, one biological replicate of the ΔliaS mutant was excluded based on DNA contamination of the RNA sample. The experiments were not randomized. The Investigators were not blinded to allocation during experiments and outcome assessment.

## Reporting summary
Further information on research design is available in the Nature Portfolio Reporting Summary linked to this article.

## Data availability

The CRISPRi-seq and RNA-seq data generated in this study have been deposited in the Sequence Read Archive under BioProject accession code PRJNA1185710. The source data generated in this study are provided in the Supplementary Information and Source Data file. Source data are provided with this paper.

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

## Acknowledgements

Bevika Sewgoolam was supported by the University of Lausanne, Faculty of Biology and Medicine PhD Fellowship. Work in the Veening lab is supported by the Swiss National Science Foundation (SNSF) grants 310030_192517 (J.W.V.), 310030_200792 (J.W.V.) and NCCR 51NF40_180541 (J.W.V.). We would like to thank Mark van der Linden at the German Reference Laboratory for Streptococci for the clinical fluoroquinolone resistant pneumococcal isolates, the Lausanne Genomic Technologies Facility, University of Lausanne, Switzerland, for RNA-sequencing, Babet Rozendal for technical assistance and all members of the Veening Lab for their help, contributions, and feedback.

## Author contributions

Conceptualization: B.S., P.S.G., J.W.V.

## Competing interests

J.W.V. and F.P.B. are scientific advisory board members at i-Seq Biotechnology. The remaining authors declare no competing interests.
