## [Transparent Peer Review file · Nature Communications]

Genome-wide antibiotic-CRISPRi profiling identifies LiaR activation as a strategy to resensitize fluoroquinolone-resistant *Streptococcus pneumoniae*

Corresponding Author: Professor Jan-Willem Veening

Version 0:

Reviewer comments:

Reviewer #1

(Remarks to the Author)

Overall this was a compelling narrative for understanding how new networks are associated with antibiotic susceptibility. Overall, the data supports the conclusions. There were a handful of areas that could be readily strengthened to provide a stronger mechanistic basis for their results (Major Points 1, 2, 4).

Major Points

1. The data in Figure 7 are compelling, but do not represent an ideal system for this challenge. The antibiotic exposure in the water is a constant, much unlike the real work PD/PK parameters of these drugs. Would strongly suggest demonstrating a reduction in the CFU burden (zebrafish) at a bare minimum. The real key piece of information would be a murine model of lung or blood infections whereby the synergism between these two drugs is demonstrated.
2. For the bacitracin sensitization assays, it would be useful to show an irrelevant antibiotic to demonstrate specificity of the synergy for the fluoroquinolones rather than a general sensitization to any given stressor.
3. Figure 2B, 2C, 2D, 3B, 3C, 3E, 3F, and 4 should indicate the number of independent replicates and include the error distribution for the respective curves.
4. A key question remains to what is the underlying mechanism of the enhanced sensitization. A key experiment would be to measure the fluoroquinolone uptake in the sensitive mutant (or with bacitracin) at an early point (i.e. 30 minutes) to determine whether this pathway impacts the actual uptake of the respective antibiotics which can be readily measured by mass spec.

Minor Points

1. The statement on line 413-414 while technically correct is a bit misleading. The MIC breakpoints did not resensitize the isolate at the MIC breakpoint is still above the EUCAST resistance guidelines.
2. In Figure 2, why was levofloxacin used in panels B and C but ciprofloxacin used in panel D? Did the phenotypes cross-validate between the different antibiotics?

Reviewer #2

(Remarks to the Author)

The manuscript by Sewoolam et al. describes the effort to understand the genome-wide factors that contribute to antibiotic susceptibility and inform the development of new treatments for pneumococcal infections, particularly the use of fluoroquinolones on *S. pneumoniae*. The premise of this study centers around the rise of antibiotic-resistant bacterial infections and the insufficient treatments to combat these infections. The study team proposes the re-purposing of important antibiotics for new treatment use or as co-therapeutics may have a significant role in antibiotic-resistant pneumococcal infections. The authors utilized an innovative CRISPRi-Seq to determine the gene fitness of *S. pneumoniae* under fluoroquinolone-induced stress. 44 operons were found to be differentially essential across three fluoroquinolones, causing either a fitness disadvantage or advantage for the bacteria. The authors further validated 21 operons phenotypically through

the construction of individual CRISPRi knockdown strains, confirming the increased or decreased sensitivity in the majority of the strains. Of interest to the authors, the screen identified the LiaFSR operon as important under fluoroquinolone-induced stress. This operon is a two-component regulatory system that responds to cell envelope stress and is generally induced by cell-wall targeting antibiotics. The authors found that the upregulation of the response regulator LiaR regulon through either deletion of the histidine kinase LiaS or constitutive phosphorylation of LiaR, caused hypersensitivity to ciprofloxacin. Through the understanding of LiaFSR regulation, the authors sought to test a synergistic combination therapy of bacitracin combined with either levofloxacin or ciprofloxacin. Bacitracin was found to be synergistic in combination with levofloxacin, rendering previously levofloxacin resistant clinical *S. pneumoniae* strains sensitive in vitro. The authors also used an in vivo zebrafish embryo pneumococcal meningitis model to determine if the bacitracin/levofloxacin combined therapy acts synergistically. The combined treatment did increase survival, leading the authors to conclude that the combined treatment acts synergistically and bacitracin can potentiate levofloxacin to more effectively treat fluoroquinolone-resistant pneumococci.

Overall the manuscript is clearly written, organized, and utilizes outstanding experimental design in probing genome-wide gene signatures in response to antibiotic-induced stress. Given the challenges we will continue to face in combating antibiotic resistant bacteria, the present study offers a potential pathway for the treatment of resistant *S. pneumoniae*. Flaws are relatively minor but will be important to clarify for readers.

Specific Comments:

1. Results, line 165: "Based on the CRISPRi-seq data, 21 operons of some of the top hits were selected for phenotypic validation." The authors could provide a brief explanation or rationale as to why the specific hits were chosen for validation over the other 23.
2. Results, Figure 3A: The authors have provided a visual of the LiaFSR operon organization. However, this model could be better visualized in its genomic context.
3. Results, Figure 3F: The colors cannot be fully visualized (where are the black solid/dashed lines?).
4. Results, lines 338-339: "This further confirmed. . ." No direct evidence of LiaR phosphorylation (e.g., Phos-Tag) is provided that would confirm. Suggest rewording to "further supports".
5. Results, lines 344-348: The predicted model is purely speculative and not consistent with the literature including the stoichiometry of LiaF and LiaS in the membrane: (i) most HKs exist in homodimers, (ii) studies in *B. subtilis* suggest that LiaF exists in excess relative to LiaS (PMID 23279150), (iii) are there data to suggest that HKs "bind" RRs as a mechanism of inactivation? Given the large degree of speculation here, suggest removal of Figure 5 and associated text as it detracts from the overall story.
6. Discussion, lines 480-481: "Upon the detection of membrane stress. . ." While the data presented support LiaR phosphorylation in the absence of LiaS, this is not consistent with other LiaFSR systems where LiaF appears to play the primary role as an inhibitor or activator of LiaS that then dephosphorylates or phosphorylates (respectively) LiaR. As suggested, primary LiaR phosphorylation would be derived from intracellular acetyl phosphate and that is not what is suggested in "similar models have been proposed (line 486)" (PMID 23279150; reference 59) where LiaR is phosphorylated in the setting of a *liaS::kan* disruption. Given the absence of data in how LiaFSR functions in *S. pneumoniae* (and is not the focus of the manuscript), the authors would be better served to simply describe what their data supports (LiaR phosphorylation in the absence of LiaS) and how that differs from the known literature. Clearly, LiaFSR is in need of more investigation and not an emphasis of this manuscript.
7. Discussion, lines 505-508: The authors state "combining BAC with LEV could reduce the necessary administered doses required, potentially minimizing the side effects and making systemic use of BAC clinically relevant. Additionally, investigations into the design of BAC analogues with improve systemic safety are also currently underway". The authors are providing an elusive explanation of the use of bacitracin in the clinical setting. BAC is not used intravenously (in humans) due to the high toxicity (nephrotoxicity and localized reactions) and is not orally absorbed. This section would be better served to describe ways to activate the LiaFSR system through drug discovery/HTS or through AMP/AMP analogues as adjunctive therapies. There is a large degree of unknown in how LiaFSR is activated in *S. pneumoniae* relative to other Gram-positives. Would daptomycin work better (as in *Enterococcus*) – includes the benefit of a currently approved antimicrobial for Gram-positives.

Reviewer #3

(Remarks to the Author)

The manuscript "Genome-wide antibiotic-CRISPRi profiling identifies LiaR activation as a strategy to resensitize fluoroquinolone-resistant *Streptococcus pneumoniae*" by Sewgoolam et al. describes a CRISPRi-based screen for the identification of genes that affect the susceptibility of pneumococci toward fluoroquinolone antibiotics. As a main hit, the authors identified the LiaFSR regulatory system which has previously shown to be associated with cell membrane stress and cell wall-targeting antibiotics. Interestingly, activation of the LiaR-regulon by bacitracin leads to improved killing of *S. pneumoniae* when combined with fluoroquinolone antibiotics. All experiments were conducted carefully and include proper controls. In addition, the paper is well-written and provides a solid background about the problem of antibiotic-resistant streptococci. In toto, I believe that this study is of interest to microbiologists, clinicians, and the general scientific community. However, I have spotted a few issues the authors should address in a revised version (see below).

1. It is not clear how exactly the authors verified that each individual *S. pneumoniae* mutant derived from the CRISPRi library

bears a single mutation only. This should be clarified.

2. Although it is appreciated that the authors included a zebrafish embryo model, the chosen in vivo model is debatable. To demonstrate synergism between bacitracin and fluoroquinolone antibiotics, it is recommended to use a more physiological relevant model of pneumonia (mice). This would also allow the authors to examine bacterial loads in lung tissues, thereby providing robust information on the efficacy of the identified drug combinations.

3. It is not clear why Lev has been added to the water instead of using injection (zebrafish embryo model). How can the authors ensure that Lev penetrates the embryo in sufficient concentrations?

4. Although the drug-drug interaction assay (Fig. 6A) nicely demonstrates the synergistic effect of Cip and Bac, Bac alone already kills large proportions of the bacterial populations (no uniform lawn). This experiment should be repeated by using a Bac concentration that does not inhibit streptococcal growth when applied alone.

5. While the authors have provided details on the pneumococcal strains used, the authors did not list the exact *S. aureus* and *B. subtilis* strain names. Please add this information. In this regard, I feel that at least some of the in vitro assays should be repeated with a few more isolates to exclude strain-specific effects.

Version 1:

Reviewer comments:

Reviewer #1

(Remarks to the Author)

Overall the authors did a reasonable job of responding to previous comments. My previous concern with regards to the use of the zebrafish as opposed to the murine model has not been addressed- while this would be challenging to do, this could be done in my opinion. If this is impossible due to toxicity, it calls into question the significance of the work as there is no application/real world relevance of these findings.

Reviewer #2

(Remarks to the Author)

The authors provided additional clarity in their rationale in choosing specific CRISPRi-Seq hits for validation, as well as amending the manuscript discussion to better highlight the need for drug discovery efforts to identify LiaFSR-activating compounds. They have also revised minor rewording throughout the manuscript. The authors provided additional context related to the LiaFSR operon organization model i.e. including genes up and downstream of the operon. In regard to the speculative LiaFSR model, I am satisfied with the revision of the speculative LiaFSR model (Figure 5). While this model is not consistent with the literature and most other proposed LiaFSR models, the authors have provided additional context in the figure legend and cited relative literature that shows similarities of this proposed LiaFSR system with one in another bacterial species.

Reviewer #3

(Remarks to the Author)

In the revised version of their manuscript, Sewgoolam et al. addressed some of the raised concerns. While most of the issues have been clarified, the lack of an appropriate mammalian in vivo model represents a limitation of this study and should still be addressed. In this regard, it is worth noting that bacitracin is used in animal studies as well as in clinical settings. The synergistic effect of bacitracin/levofloxacin proposed by the authors could even contribute to a significant reduction in bacitracin-induced nephrotoxicity. Experiments that address this issue would elevate the paper and may help to expand the potential applications of bacitracin in clinics.

We thank the reviewers for their positive opinion on our work and sincerely appreciate the reviewers constructive feedback and valuable suggestions. The revisions made in response to the comments have certainly improved the manuscript. Below, we provide detailed point-by-point responses to the feedback and outline the revisions that have been made to the manuscript.

Reviewer #1 (Remarks to the Author)

Overall this was a compelling narrative for understanding how new networks are associated with antibiotic susceptibility. Overall, the data supports the conclusions. There were a handful of areas that could be readily strengthened to provide a stronger mechanistic basis for their results (Major Points 1, 2, 4).

Major Points

1. The data in Figure 7 are compelling, but do not represent an ideal system for this challenge. The antibiotic exposure in the water is a constant, much unlike the real world PD/PK parameters of these drugs. Would strongly suggest demonstrating a reduction in the CFU burden (zebrafish) at a bare minimum. The real key piece of information would be a murine model of lung or blood infections whereby the synergism between these two drugs is demonstrated.

We thank the reviewer for the valuable suggestion regarding the use of a murine model. While we acknowledge that a mammalian model would be of interest, we believe it is not the most suitable choice at this stage of our study. The zebrafish model was selected to demonstrate initial proof-of-concept of synergy. Importantly, bacitracin is not used systemically in clinical settings due to its nephrotoxicity. Conducting studies in a murine model would therefore be more appropriate once safer bacitracin analogues become available – such alternatives are currently under development (Buijs et al., 2024). We have incorporated this point into the discussion.

The use of the murine model at this stage also presents ethical considerations as optimizing a safe dosage of bacitracin will require extensive testing and a significant number of animals.

Regarding the reduction in CFU burden in the zebrafish model, this is an excellent point and has now been tested and included as new **Supp Fig 9**. The survival was directly correlated with a reduction in CFU.

2. For the bacitracin sensitization assays, it would be useful to show an irrelevant antibiotic to demonstrate specificity of the synergy for the fluoroquinolones rather than a general sensitization to any given stressor.

Good point. Checkerboard assays have been performed and added to the supplementary figures demonstrating synergy is specific to the fluoroquinolone and bacitracin combination (**Supp Fig 11**).

3. Figure 2B, 2C, 2D, 3B, 3C, 3E, 3F, and 4 should indicate the number of independent replicates and include the error distribution for the respective curves.

The figure legends in the revised manuscript now indicate that all growth curves have been conducted in biological triplicate and the error distribution has been added.

4. A key question remains to what is the underlying mechanism of the enhanced sensitization. A key experiment would be to measure the fluoroquinolone uptake in the sensitive mutant (or with bacitracin)

at an early point (i.e. 30 minutes) to determine whether this pathway impacts the actual uptake of the respective antibiotics which can be readily measured by mass spec.

This is a great question and suggestion, and we propose to do this in future work by metabolomics and in fact have already started some experimental pilots. However, it will take some serious time and effort by us and our metabolomics core facility to get this data. But this will certainly be part of a follow up study.

Minor Points

1. The statement on line 413-414 while technically correct is a bit misleading. The MIC breakpoints did not resensitize the isolate at the MIC breakpoint is still above the EUCAST resistance guidelines.

This statement has been rephrased (L.358 now).

2. In Figure 2, why was levofloxacin used in panels B and C but ciprofloxacin used in panel D? Did the phenotypes cross-validate between the different antibiotics?

Indeed, the phenotypes cross-validate between levofloxacin and ciprofloxacin. However, for consistency within Figure 2, we have changed the figure so that all panels use levofloxacin.

Reviewer #2 (Remarks to the Author)

The manuscript by Sewoolam et al. describes the effort to understand the genome-wide factors that contribute to antibiotic susceptibility and inform the development of new treatments for pneumococcal infections, particularly the use of fluoroquinolones on *S. pneumoniae*. The premise of this study centers around the rise of antibiotic-resistant bacterial infections and the insufficient treatments to combat these infections. The study team proposes the re-purposing of important antibiotics for new treatment use or as co-therapeutics may have a significant role in antibiotic-resistant pneumococcal infections. The authors utilized an innovative CRISPRi-Seq to determine the gene fitness of *S. pneumoniae* under fluoroquinolone-induced stress. 44 operons were found to be differentially essential across three fluoroquinolones, causing either a fitness disadvantage or advantage for the bacteria. The authors further validated 21 operons phenotypically through the construction of individual CRISPRi knockdown strains, confirming the increased or decreased sensitivity in the majority of the strains. Of interest to the authors, the screen identified the LiaFSR operon as important under fluoroquinolone-induced stress. This operon is a two-component regulatory system that responds to cell envelope stress and is generally induced by cell-wall targeting antibiotics. The authors found that the upregulation of the response regulator LiaR regulon through either deletion of the histidine kinase LiaS or constitutive phosphorylation of LiaR, caused hypersensitivity to ciprofloxacin. Through the understanding of LiaFSR regulation, the authors sought to test a synergistic combination therapy of bacitracin combined with either levofloxacin or ciprofloxacin. Bacitracin was found to be synergistic in combination with levofloxacin, rendering previously levofloxacin resistant clinical *S. pneumoniae* strains sensitive in vitro. The authors also used an in vivo zebrafish embryo pneumococcal meningitis model to determine if the bacitracin/levofloxacin combined therapy acts synergistically. The combined treatment did increase survival, leading the authors to conclude that the combined treatment acts synergistically and bacitracin can potentiate levofloxacin to more effectively treat fluoroquinolone-resistant pneumococci.

Overall the manuscript is clearly written, organized, and utilizes outstanding experimental design in probing genome-wide gene signatures in response to antibiotic-induced stress. Given the challenges we will continue to face in combating antibiotic resistant bacteria, the present study offers a potential pathway for the treatment of resistant *S. pneumoniae*. Flaws are relatively minor but will be important to clarify for readers.

Specific Comments:

1. Results, line 165: “Based on the CRISPRi-seq data, 21 operons of some of the top hits were selected for phenotypic validation.” The authors could provide a brief explanation or rationale as to why the specific hits were chosen for validation over the other 23.

A sentence regarding the selection of the genes has been included (L.166-169).

2. Results, Figure 3A: The authors have provided a visual of the LiaFSR operon organization. However, this model could be better visualized in its genomic context.

The model has been edited to include genes up and downstream of *liaFSR*.

3. Results, Figure 3F: The colors cannot be fully visualized (where are the black solid/dashed lines?).

The figure legend has been edited here for clarity.

4. Results, lines 338-339: “This further confirmed. . .” No direct evidence of LiaR phosphorylation (e.g., Phos-Tag) is provided that would confirm. Suggest rewording to “further supports”.

This has been rephrased.

5. Results, lines 344-348: The predicted model is purely speculative and not consistent with the literature including the stoichiometry of LiaF and LiaS in the membrane: (i) most HKs exist in homodimers, (ii) studies in *B. subtilis* suggest that LiaF exists in excess relative to LiaS (PMID 23279150), (iii) are there data to suggest that HKs “bind” RRs as a mechanism of inactivation? Given the large degree of speculation here, suggest removal of Figure 5 and associated text as it detracts from the overall story.

6. Discussion, lines 480-481: “Upon the detection of membrane stress. . .” While the data presented support LiaR phosphorylation in the absence of LiaS, this is not consistent with other LiaFSR systems where LiaF appears to play the primary role as an inhibitor or activator of LiaS that then dephosphorylates or phosphorylates (respectively) LiaR. As suggested, primary LiaR phosphorylation would be derived from intracellular acetyl phosphate and that is not what is suggested in “similar models have been proposed (line 486)” (PMID 23279150; reference 59) where LiaR is phosphorylated in the setting of a *liaS::kan* disruption. Given the absence of data in how LiaFSR functions in *S. pneumoniae* (and is not the focus of the manuscript), the authors would be better served to simply describe what their data supports (LiaR phosphorylation in the absence of LiaS) and how that differs from the known literature. Clearly, LiaFSR is in need of more investigation and not an emphasis of this manuscript.

We appreciate the reviewer’s comments regarding the stoichiometry, number of molecules and dimers in the model. This is however a speculative/hypothetical model, and we have now clarified this in the figure legend. We have also mentioned specifically these caveats regarding the model and cited the relevant literature. Our genetic data does indeed show that LiaS is not acting as a histidine kinase and AlphaFold modeling provides additional support to the model by demonstrating that these proteins interact and form a complex. We also furthered clarified that while this model is different from the model put forward for *B. subtilis*, it matches the model previously proposed for *Listeria monocytogenes* (Fritsch et al., 2011).

While we acknowledge that further investigation is needed, we believe the model remains valuable and contributes to the literature by serving as a base model for future experimental validation.

7. Discussion, lines 505-508: The authors state “combining BAC with LEV could reduce the necessary administered doses required, potentially minimizing the side effects and making systemic use of BAC clinically relevant. Additionally, investigations into the design of BAC analogues with improve systemic safety are also currently underway”. The authors are providing an elusive explanation of the use of bacitracin in the clinical setting. BAC is not used intravenously (in humans) due to the high toxicity (nephrotoxicity and localized reactions) and is not orally absorbed. This section would be better served to describe ways to activate the LiaFSR system through drug discovery/HTS or through AMP/AMP analogues as adjunctive therapies. There is a large degree of unknown in how LiaFSR is activated in *S. pneumoniae* relative to other Gram-positives. Would daptomycin work better (as in *Enterococcus*) – includes the benefit of a currently approved antimicrobial for Gram-positives.

The reviewer is correct here regarding the clinical use of bacitracin. We have tested daptomycin along with several other membrane-disrupting compounds (e.g. polymyxin B, colistin, vancomycin), however in our hands these compounds did not strongly induce the pneumococcal LiaR regulon and subsequently exhibit synergy with the fluoroquinolones.

We have revised the discussion to highlight the need for further drug discovery efforts to identify compounds similar to bacitracin/AMPs/ more compounds that activate LiaFSR system in *S. pneumoniae*. (L.442-445).

Reviewer #3

The manuscript “Genome-wide antibiotic-CRISPRi profiling identifies LiaR activation as a strategy to resensitize fluoroquinolone-resistant *Streptococcus pneumoniae*” by Sewgoolam et al. describes a CRISPRi-based screen for the identification of genes that affect the susceptibility of pneumococci toward fluoroquinolone antibiotics. As a main hit, the authors identified the LiaFSR regulatory system which has previously shown to be associated with cell membrane stress and cell wall-targeting antibiotics. Interestingly, activation of the LiaR-regulon by bacitracin leads to improved killing of *S. pneumoniae* when combined with fluoroquinolone antibiotics. All experiments were conducted carefully and include proper controls. In addition, the paper is well-written and provides a solid background about the problem of antibiotic-resistant streptococci. In toto, I believe that this study is of interest to microbiologists, clinicians, and the general scientific community. However, I have spotted a few issues the authors should address in a revised version (see below).

1. It is not clear how exactly the authors verified that each individual *S. pneumoniae* mutant derived from the CRISPRi library bears a single mutation only. This should be clarified.

The methodology for the generation of the CRISPRi library has been described and verified in previous work from our lab (Liu et al., 2017; de Bakker et al., 2022). Briefly, each plasmid in the CRISPRi library carries a single sgRNA which is transformed into *S. pneumoniae* and integrates at a non-essential site into the chromosome (the ZIP locus). Thus, since it is only possible to have a single integration event, each cell can only harbor and express a single unique sgRNA. If a second sgRNA construct were to also integrate, it would simply replace the first sgRNA. Thus, each clone only has a one sgRNA per cell resulting in a single operon knockdown. This has been clarified in the description of the library (L.131).

2. Although it is appreciated that the authors included a zebrafish embryo model, the chosen in vivo model is debatable. To demonstrate synergism between bacitracin and fluoroquinolone antibiotics, it is recommended to use a more physiological relevant model of pneumonia (mice). This would also allow the authors to examine bacterial loads in lung tissues, thereby providing robust information on the efficacy of the identified drug combinations.

As discussed above to Reviewer#1:

We thank the reviewer for the valuable suggestion regarding the use of a murine model. While we acknowledge that a mammalian model would be of interest, we believe it is not the most suitable choice at this stage of our study. The zebrafish model was selected to demonstrate initial proof-of-concept of synergy. Importantly, bacitracin is not used systemically in clinical settings due to its nephrotoxicity. Conducting studies in a murine model would therefore be more appropriate once safer bacitracin analogues become available – such alternatives are currently under development (Buijs et al., 2024). We have incorporated this point into the discussion.

The use of the murine model at this stage also presents ethical considerations as optimizing a safe dosage of bacitracin will require extensive testing and a significant number of animals.

We have additionally included bacterial load data (CFU per zebrafish embryo) in **Supp Fig 9**.

3. It is not clear why Lev has been added to the water instead of using injection (zebrafish embryo model). How can the authors ensure that Lev penetrates the embryo in sufficient concentrations?

We added LEV to the water as it is most commonly used as an oral drug in the treatment of patients with infectious diseases and has excellent tissue penetration characteristics so there was no need to directly inject it. Moreover, the observation that zebrafish embryos treated with LEV alone showed increased survival suggests that the drug is absorbed at a sufficient concentration. Additionally previous studies have demonstrated that small molecules can be absorbed by zebrafish through the water (Patton et al., 2021). In contrast, bacitracin was administered via injection as it is typically used intramuscularly or topically in humans and has a relatively large molecular size which limits its uptake from the water.

4. Although the drug-drug interaction assay (Fig. 6A) nicely demonstrates the synergistic effect of Cip and Bac, Bac alone already kills large proportions of the bacterial populations (no uniform lawn). This experiment should be repeated by using a Bac concentration that does not inhibit streptococcal growth when applied alone.

We conducted experiments using lower concentrations of BAC and the synergy is consistent between concentrations. Figure included below.

5. While the authors have provided details on the pneumococcal strains used, the authors did not list the exact *S. aureus* and *B. subtilis* strain names. Please add this information. In this regard, I feel that at least some of the in vitro assays should be repeated with a few more isolates to exclude strain-specific effects.

This is a good point. We have now performed additional disc diffusion assays using additional *S.aureus* and *B. subtilis* strains (new **Supp Fig 8**) and we have included the information on the strains used.

References:

1. Buijs, N. P. *et al.* A classic antibiotic reimaged: Rationally designed bacitracin variants exhibit potent activity against vancomycin-resistant pathogens. *Proc. Natl. Acad. Sci. U.S.A.* **121**, e2315310121 (2024).
2. Fritsch, F. *et al.* The cell envelope stress response mediated by the LiaFSR Lm three-component system of *Listeria monocytogenes* is controlled via the phosphatase activity of the bifunctional histidine kinase LiaS Lm. *Microbiology* **157**, 373–386 (2011).
3. Liu, X. *et al.* High-throughput CRISPRi phenotyping identifies new essential genes in *Streptococcus pneumoniae*. *Molecular Systems Biology* **13**, 931 (2017). 13(5):931.
4. De Bakker, V., Liu, X., Bravo, A. M. & Veening, J.-W. CRISPRi-seq for genome-wide fitness quantification in bacteria. *Nat Protoc* **17**, 252–281 (2022).
5. Patton, E. E., Zon, L. I., & Langenau, D. M. Zebrafish disease models in drug discovery: from preclinical modelling to clinical trials. *Nature reviews. Drug discovery*, 20(8), 611–628.] (2021).

REVIEWERS' COMMENTS

We thank the reviewers for their work and sincerely appreciate the reviewers constructive feedback and suggestions.

Reviewer #1 (Remarks to the Author):

Overall the authors did a reasonable job of responding to previous comments. My previous concern with regards to the use of the zebrafish as opposed to the murine model has not been addressed- while this would be challenging to do, this could be done in my opinion. If this is impossible due to toxicity, it calls into question the significance of the work as there is no application/real world relevance of these findings.

Reviewer #2 (Remarks to the Author):

The authors provided additional clarity in their rationale in choosing specific CRISPRi-Seq hits for validation, as well as amending the manuscript discussion to better highlight the need for drug discovery efforts to identify LiaFSR-activating compounds. They have also revised minor rewording throughout the manuscript. The authors provided additional context related to the LiaFSR operon organization model i.e. including genes up and downstream of the operon. In regard to the speculative LiaFSR model, I am satisfied with the revision of the speculative LiaFSR model (Figure 5). While this model is not consistent with the literature and most other proposed LiaFSR models, the authors have provided additional context in the figure legend and cited relative literature that shows similarities of this proposed LiaFSR system with one in another bacterial species.

Reviewer #3 (Remarks to the Author):

In the revised version of their manuscript, Sewgoolam et al. addressed some of the raised concerns. While most of the issues have been clarified, the lack of an appropriate mammalian in vivo model represents a limitation of this study and should still be addressed. In this regard, it is worth noting that bacitracin is used in animal studies as well as in clinical settings. The synergistic effect of bacitracin/levofloxacin proposed by the authors could even contribute to a significant reduction in bacitracin-induced nephrotoxicity. Experiments that address this issue would elevate the paper and may help to expand the potential applications of bacitracin in clinics.